# Uncovering the viral aetiology of undiagnosed acute febrile illness in Uganda using metagenomic sequencing

Shirin Ashraf[1,8], Hanna Jerome[1,8], Daniel Lule Bugembe[2,3], Deogratius Ssemwanga[2,3], Timothy Byaruhanga[2,6], John Timothy Kayiwa[2], Robert Downing [2], Jesus F. Salazar-Gonzalez [3,7], Maria G. Salazar[3], James G. Shepherd [1], Craig Wilkie [4], Chris Davis [1], Nicola Logan[1], Sreenu B. Vattipally [1], Gavin S. Wilkie[1], Ana da Silva Filipe [1], Alfred Ssekagiri[1,2], Prossy Namuwulya[1,2], Henry Bukenya[2], Brian K. Kigozi[2], Weronika Witkowska McConnell[1], Brian J. Willett [1], Stephen Balinandi[2], Julius Lutwama[2], Pontiano Kaleebu[2,5], Josephine Bwogi[2,9] ✉ & Emma C. Thomson [1,5,9] ✉

Viruses associated with acute febrile illness in Africa cause a spectrum of clinical disease from mild to life-threatening. Routine diagnostic methods are insufficient to identify all viral pathogens in this region. In this study, 1281 febrile Ugandan patients were prospectively recruited as part of the CDC-UVRI Acute Febrile Illness Study and pre-screened for common pathogens. 210/1281 undiagnosed samples, and 20 additional samples from viral outbreaks were subjected to metagenomic sequencing. Viral pathogens were identified in 44/230 (19%), including respiratory, hepatitis, blood-borne, gastrointestinal and vector-borne viruses. Importantly, one case of Crimean-Congo haemorrhagic fever and two cases each of Rift Valley fever, dengue and yellow fever were detected in 7/230 (3%) of cases. Le Dantec virus, last reported in 1969, was also identified in one patient. The presence of high-consequence and (re-)emerging viruses of public health concern highlights the need for enhanced population-based diagnostic surveillance in the African region.

East Africa is a highly biodiverse region and a hotspot for viral zoonoses. As a result of relative political stability and excellent infrastructure, Uganda is well-placed as a sentinel location for virus surveillance. In the last decade, Uganda has detected outbreaks of haemorrhagic fever caused by Ebola virus (EBOV), Sudan virus (SUDV), Marburg virus (MARV), Rift Valley fever virus (RVFV), yellow fever virus (YFV) and Crimean-Congo haemorrhagic fever virus (CCHFV)[1]. Many vector-borne viral infections are also prevalent, and several were described here or in neighbouring countries for the first time, including Zika virus (ZIKV), Semliki Forest virus (SFV), chikungunya virus (CHIKV), O'nyong nyong virus (ONNV) and West Nile virus (WNV)[2]. Anthropogenic change, including expanding travel networks,

[1]MRC-University of Glasgow Centre for Virus Research, (CVR), Glasgow, UK. [2]Uganda Virus Research Institute, (UVRI), Entebbe, Uganda. [3]MRC/UVRI & LSHTM Uganda Research Unit, (MRC-UVRI), Entebbe, Uganda. [4]School of Mathematics and Statistics, University of Glasgow, Glasgow, UK. [5]London School of Hygiene and Tropical Medicine, (LSHTM), Glasgow, UK. [6]Present address: Department of Virology, Animal and Plant Health Agency, (APHA), Surrey, UK. [7]Present address: National Institute of Allergy and Infectious Diseases, National Institutes of Health, (NIH), Rockville, MD, USA. [8]These authors contributed equally: Shirin Ashraf, Hanna Jerome. [9]These authors jointly supervised this work: Josephine Bwogi, Emma C. Thomson. ✉e-mail: jbwogi@uvri.go.ug; emma.thomson@glasgow.ac.uk

and shifting global climate patterns raise the risk of spread within and outside zoonotic hotspots[3]. Many infections are missed due to a scarcity of diagnostic tests in resource-constrained settings, increasing the likelihood of the spread of high-consequence viruses. Further, viral infection may be misattributed to malaria or bacterial infection, resulting in the inappropriate use of antimicrobial therapies. In this study, we investigated the occurrence of undiagnosed viral infections as a cause of acute febrile illness (AFI), using population-based metagenomic sequencing as a public health surveillance tool.

AFI is a common cause of presentation to healthcare centres across Sub-Saharan Africa (SSA), where it accounts for substantial morbidity and productivity losses, despite a significant reduction in malaria prevalence following successful intervention campaigns[4]. The contribution of viral infection to AFI in Africa is likely to be high and under-estimated, as population studies have relied on narrow-spectrum diagnostic tests limited to selected pathogens[5–8]. A thorough understanding of the burden of viral disease is essential for implementing public health prevention measures, including vaccination, provision of personal protective equipment, limiting vector-borne transmission, the introduction of cost-effective diagnostics and treatments, and the identification of research gaps for new diagnostic and therapeutic strategies. Metagenomic next-generation sequencing (mNGS) allows for the unbiased identification of viral genomes in clinical samples, including known and novel pathogens. The use of population-based syndromic surveillance by mNGS for the identification of circulating pathogens in high-risk hotspot areas is likely to have high value in enhancing surveillance for new and emerging pathogens. The widespread implementation of NGS-based SARS-CoV-2 genomic surveillance means that the infrastructure for such an approach is now available in many African countries[9].

In this study, we used mNGS to identify viral infections in undiagnosed cases from the acute febrile illness (AFI) cohort, a joint initiative between the Uganda Virus Research Institute (UVRI) and the Centers for Disease Control (CDC)[10]. In this cohort, pathogen-specific diagnostic assays had previously been applied to detect exposure to several common infections, including malaria, chikungunya virus, rickettsial infections, typhoid fever, West Nile virus, dengue virus (DENV), and leptospirosis. However, no diagnosis was made in 16% of participants. In the second part of this study, we applied pathogen-agnostic mNGS to cases that remained undiagnosed after pathogen-selected serological and PCR-based assays in part one. In addition, we investigated risk factors and clinical predictors of viral infection in patients using prospectively collected metadata.

## Results

### Patient demographics

Within the AFI cohort of 1281 patients, the median age was 18 years (IQR 7–30 years). 717 participants (56%) were female and 564 (44%) male. 382 (30%) patients were recruited from central Uganda (Wakiso), 450 (35%) from Western Uganda (Kasese) and 449 (35%) from Northern Uganda (Arua) (Fig. 1a). The median catchment distance was 1.8 km. 46%(585/1281) of patients took medication before visiting the clinic. These were obtained from clinics/doctors ($n = 382;65\%$), from a pharmacy or drug store ($n = 172;29\%$), from family members ($n = 13;2\%$) or traditional healers ($n = 11;2\%$). Medication included anti-pyretics, analgesics, antibiotics and antimalarials.

The demographic distribution in the undiagnosed subset of 210 patients was similar, with a median age of 16 years (IQR 6–32 years). 114 participants (54%) were female and 96 (46%) were male. 87 (41%) patients were recruited in Wakiso, 73 (35%) in Kasese and 50 (24%) in

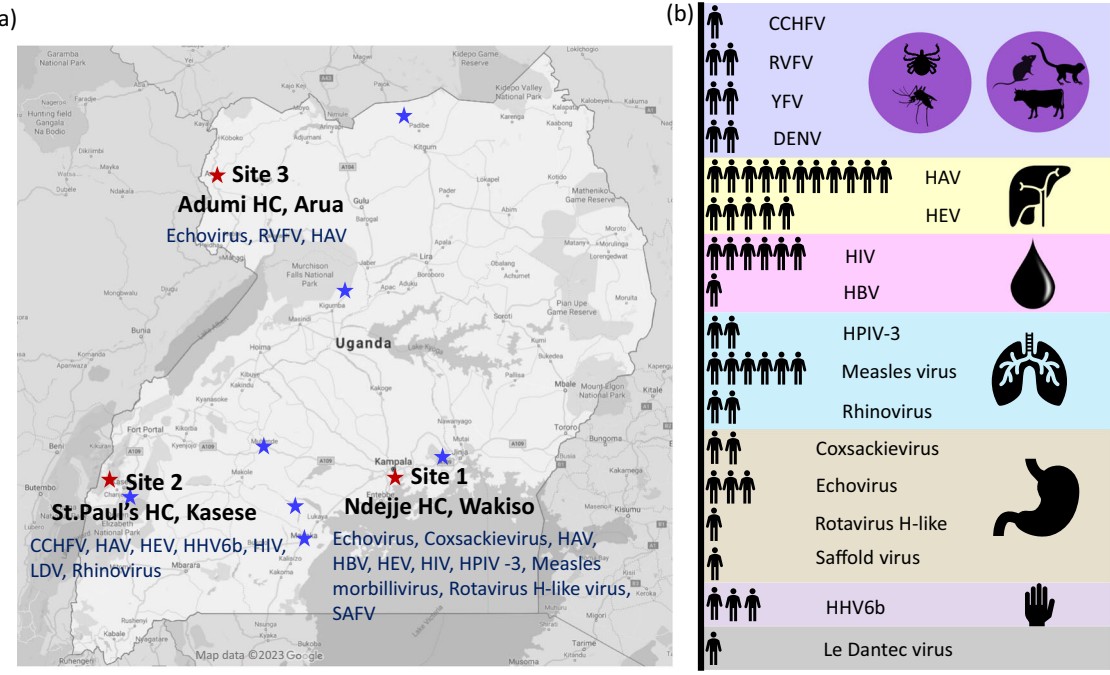

**Fig. 1 | Undiagnosed viruses identified in the AFI study. a** Map showing AFI study sites in Uganda and viruses identified from undiagnosed samples listed by site (red star). Approximate locations of outbreak samples, where available, are also marked (blue star). **b** Viruses detected by mNGS grouped as zoonoses/arboviruses (purple), hepatitis (yellow), blood-borne infections (pink), respiratory viruses (blue), gastrointestinal infections (brown), viruses associated with rash (mauve) and uncharacterised (grey). Icons show the number of patients infected with each virus. RVFV rift valley fever virus, HAV hepatitis A virus, CCHFV Crimean-Congo haemorrhagic

fever virus, HEV hepatitis E virus, HHV6b human herpesvirus-6b, HIV human immunodeficiency virus, LDV Le Dantec virus, HBV hepatitis B virus, HPIV-3 human parainfluenza virus 3, SAFV Saffold virus. Icons to the right indicate syndromic grouping/transmission routes. Species silhouettes obtained through Phylopic; tick, rodent, cattle https://creativecommons.org/publicdomain/zero/1.0/, mosquito https://creativecommons.org/publicdomain/mark/1.0/, primate (credit: Bogdan Bocianowski) https://creativecommons.org/licenses/by-sa/3.0/.

**Table 1 | Demographics and clinical characteristics of AFI patients diagnosed with viral infections using mNGS**

| n | 37 |
|---|---|
| Sex = Male/Female (%) | 23/14 (62.16/37.84) |
| Age (median [IQR]) | 7(2–40) |
| **Age bracket in years (%)** | |
| 0–5 | 16 (43.24) |
| 6–15 | 14 (37.84) |
| 16–25 | 2 (5.41) |
| 26–35 | 3 (8.11) |
| 36–45 | 2 (5.41) |
| **Study Clinic (%)** | |
| St Paul's HC | 10 (27.03) |
| Ndejje HC | 24 (64.86) |
| Adumi HC | 3 (8.11) |
| **Clinical features** | |
| Baseline temperature (mean (SD)) | 38.37 (0.92) |
| Fever duration (days) (mean (SD)) | 3.5 (1.5) |
| Rigors (%) | 8 (21.62) |
| Headache (%) | 25 (67.57) |
| Neck_pain (%) | 6 (16.22) |
| Jaundice (%) | 4 (10.81) |
| Nausea (%) | 15 (40.54) |
| Vomiting (%) | 15 (40.54) |
| Abdominal_pain (%) | 16 (43.24) |
| Diarrhoea (%) | 10 (27.03) |
| Myalgia (%) | 7 (18.92) |
| Arthralgia (%) | 11 (29.73) |
| Back_pain (%) | 7 (18.92) |
| Haemorrhage_any (%) | 3 (8.11) |
| Haemoptysis (%) | 1 (2.70) |
| Haemochezia (%) | 2 (5.41) |
| **Contact with ectoparasites (%)** | 29 (78.38) |
| Mosquitoes (%) | 28 (75.68) |
| Ticks (%) | 4 (10.81) |
| Fleas (%) | 4 (10.81) |
| Lice (%) | 1 (2.70) |
| Biting flies (%) | 4 (10.81) |
| **Health centre diagnosis (%)** | |
| Gastroenteritis | 2 (5.41) |
| Malaria | 5 (13.51) |
| Respiratory tract infection | 9 (24.32) |
| Typhoid | 22 (59.46) |
| Unknown | 2 (5.41) |
| **Antibacterial therapy (%)** | 30(81) |
| **Antimalarial therapy (%)** | 14 (37.8) |
| **Anthelmintic therapy (%)** | 5 (13.5) |

Arua. Limited clinical and demographic information was available for the outbreak samples OB1-OB20 (Supplementary Data 3).

## NGS analysis

19% of patients with undiagnosed febrile illness (44/230) had evidence of acute viral infection by mNGS (Fig. 1b), comprising 37/210 (18%) of AFI study samples and 7/20 (35%) of outbreak samples. Demographic and clinical data for the 37 AFI study samples with detected viral pathogens are shown in Table 1. The median age at clinical presentation was 7 years (range 2–40). 30/37 (81%) cases occurred in children

under the age of 16. Males were more commonly represented than females (23/37; 62%). The majority had been recently bitten by mosquitoes (28/37; 76%) and/or other arthropod vectors, including biting flies (4/37; 11%), ticks (4/37; 11%), fleas (3/37; 11%) and lice (1/37; 3%). Three undiagnosed patients (8%) subsequently found to have a viral infection reported haemorrhagic symptoms, but none had a viral haemorrhagic fever (VHF) virus detected by mNGS. At presentation, the mean temperature was 38.4 °C. Other clinical symptoms are described in Table 1. Common presumptive diagnoses included gastrointestinal infection, respiratory tract infection, typhoid fever and "unknown". 30/37 (81%) of patients were presumptively treated with antibiotics, 14 (38%) were treated with antimalarials and 5 (14%) with antihelminth therapy.

Of the 44 patients with a viral genome identified by mNGS, 7 (16%) had a virus associated with VHF (CCHFV, RVFV, YFV and DENV), 10 (23%) with viruses associated with respiratory infection (human parainfluenza virus 3, rhinovirus C and measles virus), 17 (39%) with hepatitis viruses A, B and E (HAV, HBV, HEV) and 6 (14%) with gastroenteritis-associated viruses (Table 2 and Supplementary Data 1). Six patients (17%) were infected with human immunodeficiency virus-1 (HIV-1). One patient was infected with a rhabdovirus: Le Dantec virus (LDV) and one patient was infected with Saffold virus. Three of these cases, two of yellow fever and one with RVF were correctly identified in the context of suspected outbreaks.

### Viruses associated with VHF

Genomes from four VHF-associated viruses were detected in seven febrile patients; CCHFV (patient 138-2), RVFV (patients 117-3, OB1), DENV (patient OB2, OB3) and YFV (patients OB4, OB5). None of the AFI cases had evidence of haemorrhage. Outbreak (OB) sample clinical details were not available.

The patient infected with CCHFV (138-2) was a 30-year-old male from Kasese district. He presented with a fever (37.9 °C), headache, nausea and joint pain. He had been bitten by mosquitoes but had not been aware of tick bites and had not recently slaughtered animals. No clinical diagnosis was recorded, he was treated with vitamins and paracetamol and recovered. We obtained 1,449,278 read pairs from sequencing of the serum sample. Partial (10–25% coverage) CCHFV sequences were detected by mNGS across all three segments; large (L), medium (M) and small (S) (Fig. 2a). The L and S segments had >90% nucleotide identity to previously identified genomes from Uganda and Democratic Republic of the Congo (DRC), respectively (Table 2 and Supplementary Data 1). The M segment had 81% nucleotide identity to a genome from Uganda, suggestive of reassortment (Fig. 2d). qPCR testing was equivocal with a Ct value of 41. Convalescent serology was not available for the patient, who had recovered and moved abroad.

Two cases of RVF (117-3, OB1) were detected. OB1 was suspected due to occupational risk (contact with slaughtered animals) and suggestive symptoms during an outbreak, and the diagnosis was confirmed by PCR. 117-3 was a 32-year-old woman from a rural part of Arua district who had a 3-day history of fever. She described chills, headache, eye, neck and joint pain, nausea, diarrhoea and jaundice. She had been bitten by mosquitoes and fleas. She was given paracetamol and attended a traditional healer for further treatment and did not attend for convalescent serology. Viral reads most closely matched the L segment of a Ugandan strain in case 1 (OB1) and the M segment of a Kenyan strain in case 2 (117-3).

Two cases of YFV infection and two DENV infections (genotypes 1 and 3) were detected by mNGS in samples from cases of suspected YFV infection in 2013. Yellow fever virus was detected in samples from two patients in 2013 and 2016, respectively. In each case YFV detection by mNGS was confirmed by diagnostic PCR, however, in both DENV-positive samples, in-house diagnostic PCR performed at UVRI was negative (Supplementary Data 3). Further, flavivirus IgM serology showed cross-reactivity with Zika virus (DENV IgM was negative, but

**Table 2 | Viruses detected by mNGS in undiagnosed patients in the AFI cohort**

| Virus | No. of patients | Nearest reference accession(s) | Country of nearest reference |
|---|---|---|---|
| Viruses associated with haemorrhagic fever | | | |
| CCHFV | 1 | MW464976 (L), MW464977 (M), DQ211650 (S) | Uganda (L and M) and DRC (S) |
| RVFV | 2 | MG953422, MG273456 | Uganda, Kenya |
| YFV | 2 | AY968065, JN620362 | Uganda |
| DENV | 2 | MN964273, MZ857206 | China ex Ethiopia (type 3), Kenya (type 1) |
| Viruses associated with acute hepatitis | | | |
| HAV | 11 | MH685714 | Uganda |
| HEV | 5 | MH918640 | Nigeria |
| Blood-borne viruses | | | |
| HBV | 1 | MK512473 | Rwanda |
| HIV-1 | 6 | MW006081, MW006060, AY322184, EU110093 | Uganda, Kenya |
| Respiratory viruses | | | |
| Measles B3 | 6 | ON642798, ON642800, ON642797, ON642801 | Uganda |
| Parainfluenza virus 3 | 2 | MN145875, MF973181 | China, USA |
| Rhinovirus C | 2 | JF317017, KY369882 | China, USA |
| Viruses associated with gastroenteritis | | | |
| Rotavirus H-like virus | 1 | KX362515 | Vietnam |
| Echovirus E3, E4, E9 | 3 | MF678296, KM024043, MF554740 | Australia (E25), Greece (E3), USA (E4) |
| Coxsackievirus A9, B3 | 2 | KY262575, KM201659 | China, France |
| SAFV | 1 | OP778142.1 | Uganda |
| Herpesviruses associated with rash and fever | | | |
| HHV6B | 3 | KY315537, MF511176, MF994829 | USA, Uganda, DRC |
| Rarely reported viruses associated with febrile illness | | | |
| LDV | 1 | NC_034443 | Senegal |

ZIKV IgM was positive)[11]. All DENV and YFV genomes identified by mNGS were related to East/Central African lineages (Table 2, Supplementary Data 1 and Fig. 2b, c, e, f).

## Le Dantec virus

Patient 220-2 was a 9-year-old male from Kasese district. He presented to the hospital with a 38.5 °C fever of 4 days, with headache, abdominal pain, vomiting and joint pains. He had been bitten by mosquitoes. He was treated with ciprofloxacin and paracetamol for suspected typhoid fever and made a full recovery. By mNGS, we detected the full genomic sequence of a previously rarely reported rhabdovirus; Le Dantec virus (LDV; Fig. 3a). This was confirmed by direct RT-PCR from patient plasma and Sanger sequencing. The LDV genome had 94% nucleotide identity to the RefSeq LDV genome (NC_034443.1), isolated for the first time in 1965 in Senegal and sequenced from an archival sample in 2015[12] (Fig. 3d). In keeping with an active infection, we demonstrated an immunological response to LDV glycoprotein by ELISA and pseudotype neutralisation assay, in both the patient and in a contact of the patient who had been unwell at the same time, using convalescent serum (Fig. 3b, c)[13].

## Viruses associated with acute hepatitis

Hepatitis A (HAV) and hepatitis E (HEV) viruses were detected in 11 patients (OB6, 218-1, 169-1, 184-2, 145-3, 74-1, 57-1, 79-1, 208-1, 212-1, 56-1) and 5 patients (20-2, 60-1, 355-1, 187-2, OB7) respectively. For the HAV cases, the age of infection ranged from 2 to 15 years, all were febrile at presentation and 7 reported having a headache. Patients also presented with gastrointestinal discomfort, including pain (2), constipation (2), diarrhoea (1), nausea (4), vomiting (5), anorexia (4) and one had jaundice. Two of the cases were clinically suspected as typhoid, and all patients were administered one or a combination of antibiotics, anti-malarials and paracetamol. 8/10 HAV patients reported using a public tap as a water source. 6/10 patients also reported boiling water for consumption. All HAV genomes were genotype IB and related to previously reported sequences from Uganda (Fig. 4a and Supplementary

Fig. 3e). The HEV cases were 3–27 years of age, had a fever duration of 3–5 days, and all reported a headache. Two had nausea, diarrhoea (one reported blood in stool), vomiting and anorexia. One reported a rash on their trunk, face and arms. The HEV sequences were genotype 1e and related to a Nigerian genome (Fig. 4b and Supplementary Fig. 3d). OB7 also tested positive for YFV IgM.

## Viruses transmitted predominantly by the gastrointestinal route

Viruses typically transmitted through the gastrointestinal tract (some of which may also be transmitted via the respiratory route) were detected in seven patients. Five enteroviruses were detected (the echoviruses E3, E4 and E9 and Coxsackie B3, and A9 virus) from patients 212-1, 316-3, 165-1, 195-1 and 215-1, respectively. A fragment of a highly divergent rotavirus H-like virus was detected in one patient (96–1). A genotype 3 Saffold virus was identified in patient 79-1 (Fig. 4c and Supplementary Fig. 3f) and was related (92% nucleotide identity) to a SAFV-3 genome we have previously described in a patient with measles-like illness in Uganda[14]. Two of these patients (212-1, 79-1) were co-infected with HAV, which is also transmitted by the faecal-oral route. One patient (215-1) was additionally infected with measles morbillivirus. Patients were 2–8 years old with 1–7 days of fever at presentation. Four had gastrointestinal symptoms.

## Viruses transmitted primarily via respiratory infection

Respiratory viruses were detected in plasma from ten patients and included six cases of measles morbillivirus (patients 215-1, 217-1, 242-1, 223-1, 113-1, 203-1), two cases of human parainfluenza virus 3 (patients 88-1, 114-1) and two cases of rhinovirus C (patients 40-2, OB6) (Fig. 4c, d and Supplementary Fig. 3b, c). Patient OB6 was co-infected with HAV. Measles cases ranged from age 2–9 years, all were febrile, but none were noted to have a rash. Three were clinically diagnosed with typhoid fever. The measles virus sequences closely matched Ugandan B3 genomes. The rhinovirus C clustered with subtypes RV-C5 and RV-C12.

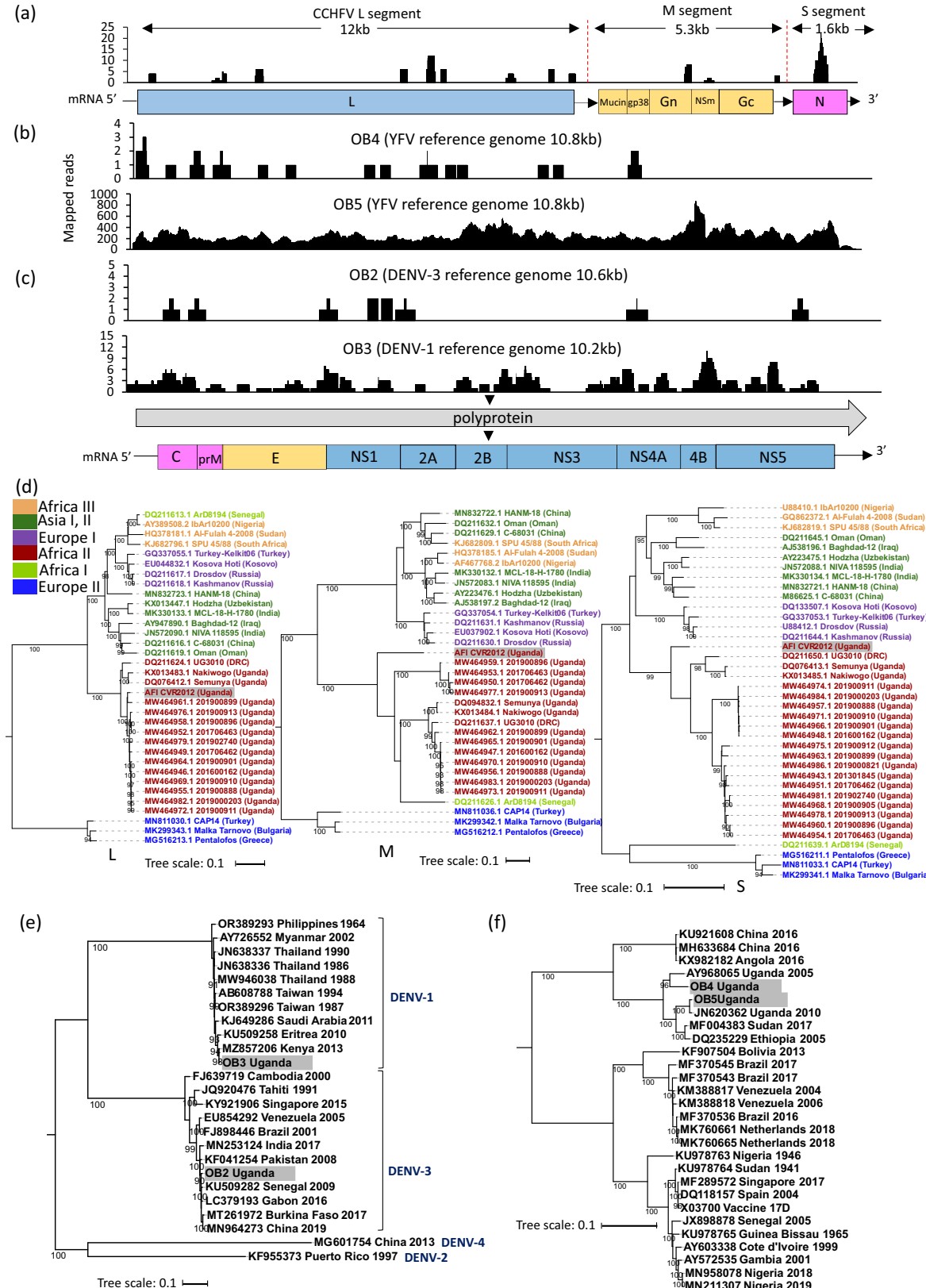

**Fig. 2 | Viruses associated with haemorrhagic fever. a–c** Coverage plots for CCHFV, YFV, DENV identified by mNGS showing number of mapped reads (y-axis) at each nucleotide position across the genome (x-axis) for each virus. Annotated reference *Orthonairovirus* and *Flavivirus* genomes indicate regions covered by sequencing reads. Genes are shown as polymerase (L) and non-structural proteins in blue, capsid and nucleoprotein in pink and surface proteins in yellow. **d** Phylogenetic tree of the full-length nucleotide sequence of CCHFV showing placement of patient isolate for L, M and S segments. Tip labels show the strain name and country of isolation. Colours indicate geographically distributed lineages of CCHFV. **e** Full genome nucleotide tree of DENV genotypes showing country and year of isolation. **f** Full genome nucleotide tree of YFV sequences showing country and year of isolation. The evolutionary scale is shown below for each tree. Ultrafast bootstrap values >90 are shown for nodes. Patient samples are highlighted in grey. Substitution models used: GTR + F + I + G4 (CCHFV-L), GTR + F + R3 (CCHFV-M), TIM2 + F + I + G4 (CCHFV-S, DENV) and GTR + F + R2 (YFV).

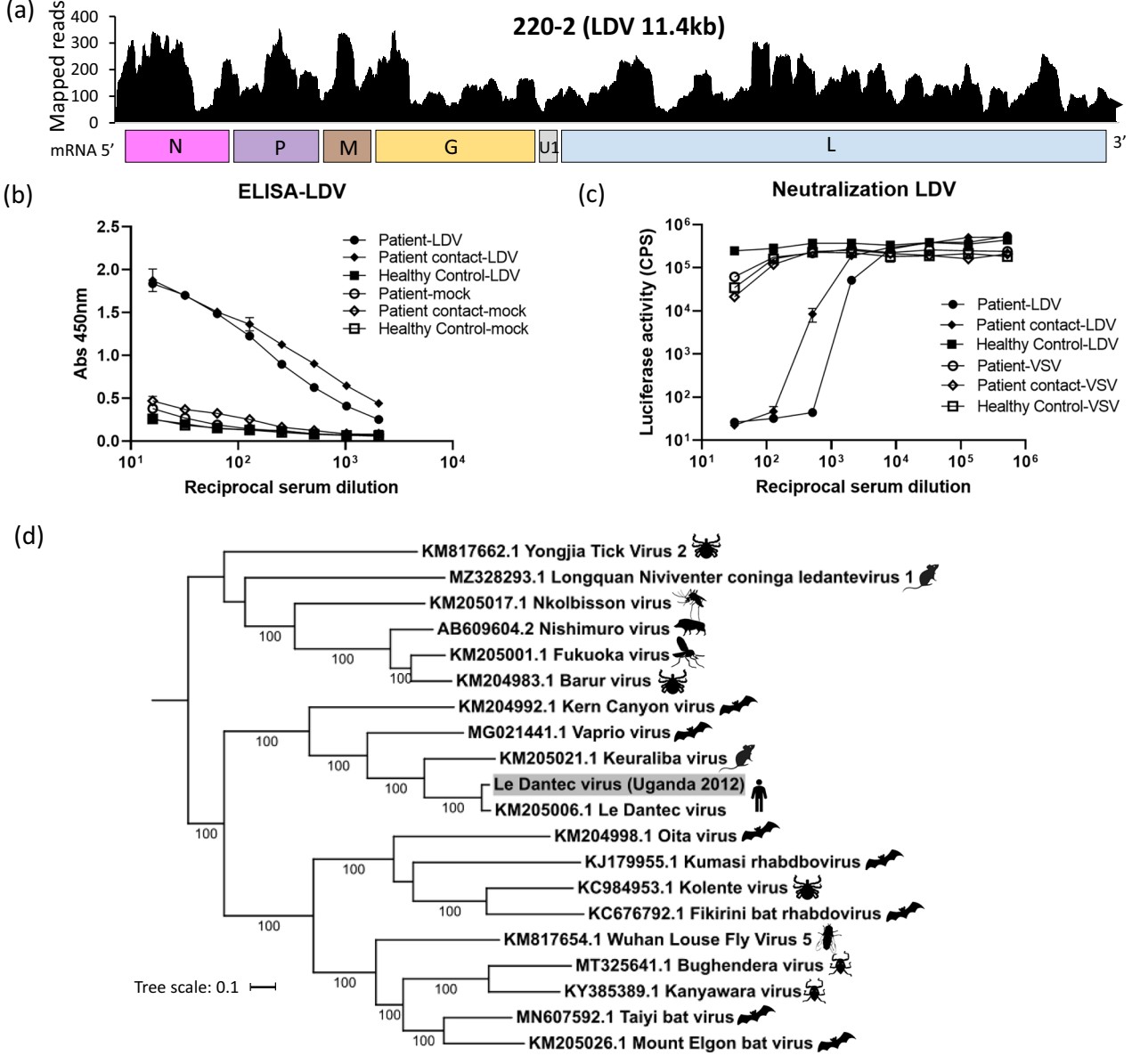

**Fig. 3 | Le Dantec virus (LDV). a** Coverage plots for LDV identified by mNGS showing number of mapped reads (y-axis) at each nucleotide position across the genome (x-axis) for each virus. The annotated reference genome below plot shows LDV genes: nucleoprotein (N), phosphoprotein (P), matrix (M), glycoprotein (G), accessory gene (U1), polymerase (L). **b** ELISA analysis showing IgG antibody response to recombinant LDV glycoprotein in patient 220-2, contact of the patient and a healthy individual from Uganda. The plasmid containing an Ephrin β2 insert was used as a mock recombinant protein. The plot shows mean absorbance OD values at 450 nm (y-axis) and standard deviation (error bars) against increasing serum dilutions (x-axis) for *n* = 3 technical replicates from one representative experiment. **c** Pseudotype neutralisation assay showing luciferase reporter activity (y-axis) against increasing serum dilutions (x-axis) for patient 220-2, contact and healthy serum using LDV pseudotype virus compared with VSV control. Data shows the mean and standard error of mean for *n* = 3 technical replicates from one representative experiment. **d** Full genome maximum-likelihood phylogenetic nucleotide tree using the GTR + F + R4 model for *Ledantevirus* sequences illustrating the reported host for each virus. Evolutionary scale and ultrafast bootstrap values >90 are shown. The patient sample is highlighted in grey. Species silhouettes obtained through Phylopic; tick, midge, boar, bat fly, rodent, louse fly https://creativecommons.org/publicdomain/zero/1.0/, mosquito https://creativecommons.org/publicdomain/mark/1.0/.

## Blood-borne viruses

The blood-borne viruses HBV (patient 192-1) and HIV-1 (patients OB1, OB5, 143-2, 390-1, 199-2 and 214-1) may have occurred either as acute or chronic infections, detected in a population with a high seroprevalence of both viruses.

HBV was detected in a 13-year-old patient (192-1) presenting with headache, anorexia and abdominal pain. It was most closely related to a sequence from Rwanda.

HIV was identified as a co-infection in three patients, one each with RVFV (OB1), YFV (OB5) and HHV6 (143-2) (Fig. 4e and Supplementary Fig. 3a); suggesting chronic infection. Patient 214-1 was infected with HIV-1 subtype A1, clustering with another Ugandan sequence. All other HIV-1 sequences were subtype D.

## Herpesviruses

Three patient samples (82-2, 185-2 and 143-2) contained human herpesvirus-6B (HHV6B) genome, the cause of roseola infantum, related to strains from the United States of America, Uganda and DRC. These may have been acute infections or the result of reactivation. Six patients (187-2, 189-2, OB8, OB9, OB10 and OB11) had evidence of

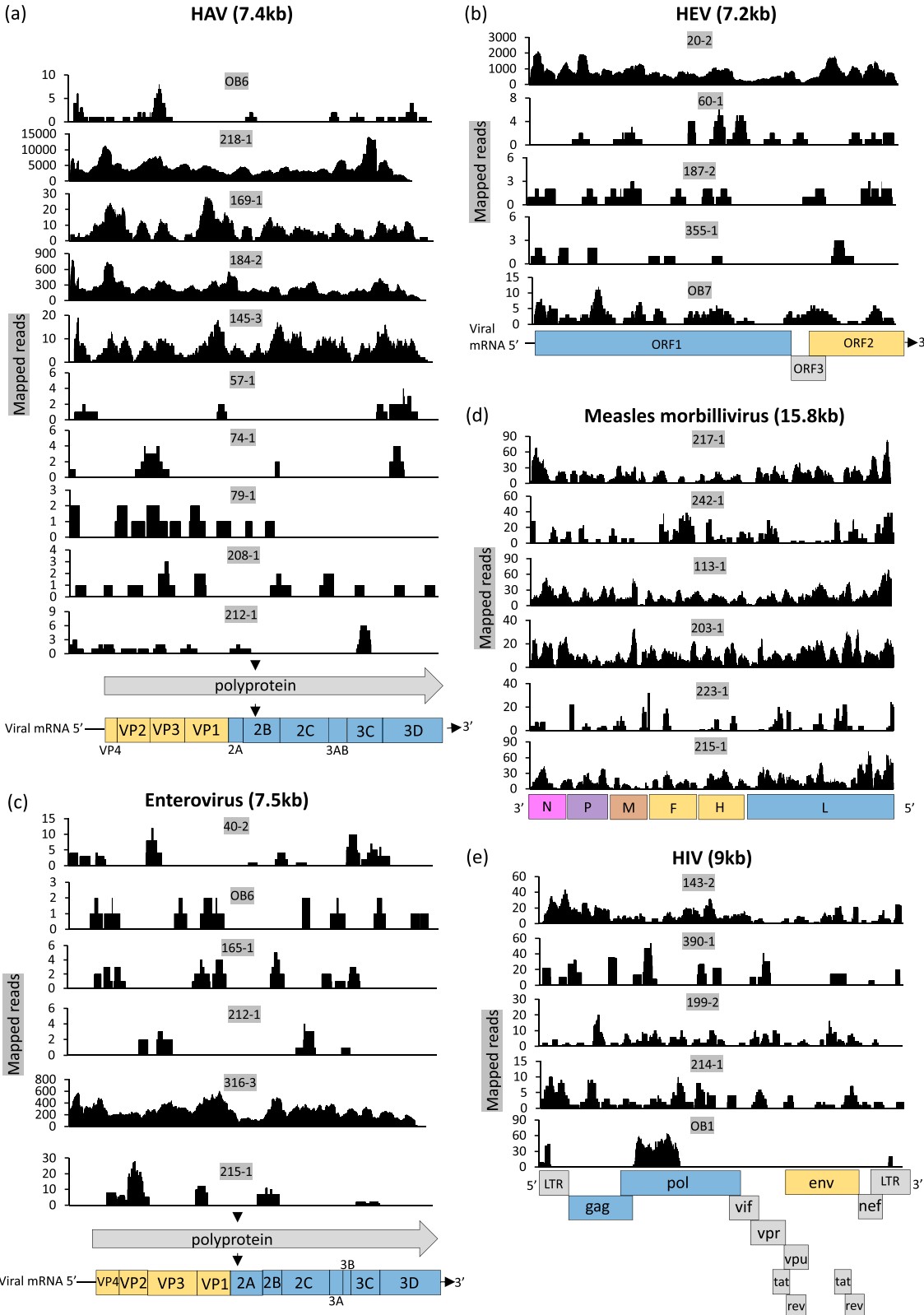

**Fig. 4 | Other viral pathogens identified from patient samples.** Coverage plots showing number of mapped reads (y-axis) against genome position (x-axis) for each patient, for **a** HAV, **b** HEV, **c** Enteroviruses including Echoviruses, Coxsackievirus and Rhinovirus C, **d** Measles virus; and **e** HIV-1. Reference genome drawings indicate depth for different regions along the sequence. Annotated reference genomes show gene positions. Structural genes/surface proteins are shown in yellow, and other genes in blue, brown, purple, pink and grey.

cytomegalovirus (HHV5) in the bloodstream; this may have been an acute or a chronic infection. No varicella-zoster virus (VZV) infections were detected.

## Viruses detected that are not known to be associated with disease

MNGS also detected viruses often found in human populations that have not been definitively associated with disease. These included the annelloviruses: Torque Teno virus (TTV), Torque Teno mini virus (TTMV), Torque Teno midi virus (TTMdV); and human pegiviruses (HPgV) (Supplementary Data 2). We also detected a second and novel rhabdovirus sequence in several AFI samples by de novo assembly (Adumi virus). This virus was likely to be a contaminant of blood collection tubes, as discussed in Supplementary Information (Supplementary Fig. 2).

## Risk factor analysis

Using the mNGS results aggregated with part I diagnostic assay results, we evaluated clinical and demographic predictors of viral infection in the whole AFI cohort (1281 patients). Stepwise multivariable analyses were performed to assess variables (demography, clinical symptoms, environment) in relation to the binary outcome of viral infection (Fig. 5a and Supplementary Data 4).

Stepwise multivariable logistic regression analysis included all variables from the univariate analysis that reached statistical significance ($p < 0.05$). In multivariable analysis, exposure to standing water was associated with nearly double the odds of viral infection (OR 1.93, 95% CI: 1.28–2.92, $p = 0.0017$). Wall spraying was a protective factor, reducing the odds of infection (OR 0.30, 95% CI: 0.10–0.91, $p = 0.033$). Using public taps was also associated with higher odds of viral infection (OR 1.45, 95% CI: 1.07–1.95, $p = 0.015$). Additionally, the use of uncovered pit latrines was associated with viral infection (OR 1.43, 95% CI: 1.05–1.95, $p = 0.022$).

Joint pain at presentation was associated with high odds of viral infection (OR 1.31, 95% CI: 1.02–1.69, $p = 0.035$). Given the high CHIKV positivity identified in the AFI cohort (523/1281 cases; 39%), and a well-described association with joint pain, a causal mediation analysis was conducted. The indirect effect of baseline joint pain on viral infection through CHIKV infection was statistically significant (ACME: 0.0906, 95% CI: 0.0171–0.15, $p = 0.010$), suggesting that CHIKV (or a related alphavirus) infection explained a significant proportion of joint pain cases.

As viruses that can cause VHF symptoms (including DENV, YFV, CCHFV and RVFV) were detected in AFI, we next aimed to analyse factors that might be associated with this presentation. Haemorrhage occurred in 88/1281 (7%) cases, and diagnoses in people with any evidence of haemorrhage (respiratory, gastrointestinal, vaginal, or petechial rash) are shown in Fig. 5b. Most VHFVs presented without haemorrhagic symptoms (one case of DENV was associated with haemorrhage). Figures 5c, d show the most common clinical diagnosis and treatment offered to patients in health centres based of availability of diagnostics and choice of treatment.

## Discussion

In this prospective cohort study of AFI in Uganda, we used mNGS to identify viral causes of acute undiagnosed fever in 210/1281 (16.4%) patients who tested negative after extensive screening for well-characterised pathogens, and an additional 20 samples obtained from suspected viral outbreaks. The cohort was prospectively recruited from patients presenting to three healthcare centres in rural and urban Uganda. We detected viral pathogens in 44/230 (19%) of undiagnosed patients, of which 7/44 (16%) were associated with VHF (two AFI and five outbreak patients). Lack of clinical diagnosis of such high-consequence infections means that onward transmission remains a risk in Uganda, and appropriate treatment may not be given. The

most commonly detected viruses reflected acute gastrointestinal and respiratory infections. The additional detection of active blood-borne virus infections reflects a chronic burden of disease and a need to widen treatment in the Ugandan population. Importantly, acute viral infection was invariably misdiagnosed as bacterial infection or malaria and treated with antibiotics or antimalarials, associated with a cost burden and a risk of emerging antimicrobial drug resistance. We also noted several vaccine-preventable infections in children (HAV, HBV, measles, and rotavirus), indicating the need to consider the implementation of a more comprehensive vaccination schedule in Uganda. Finally, we assessed clinical and environmental predictors of viral infection for the larger AFI study of 1281 patients, highlighting a significant arbovirus infection burden.

AFI is widespread in SSA, and in the absence of specialised point-of-care diagnostics; clinical symptoms associated with different pathogens may be indistinguishable. Even high-consequence VHFs such as Lassa fever and Ebola virus disease present with haemorrhage in less than 20% of cases[15,16]. We identified CCHFV in a patient with undifferentiated fever who did not report contact with ticks or slaughtered animals, highlighting the limitations of current risk algorithms in identifying such patients in an endemic area. We have recently reported a high CCHFV seroprevalence in farming communities and domesticated animals in Uganda, suggesting that unrecognised infection is common[17].

We also detected the mosquito-borne infections RVFV, DENV and YFV; of which RVF can also be caused by contact with infected livestock. RVF is widespread across Africa and may cause a spectrum of illness from mild to severe. It is associated with haemorrhage and encephalitis in around 1% of human cases[18]. DENV is endemic in Asia, South America and sub-Saharan Africa. It is relatively less well reported from the African continent, although studies suggest a high seroprevalence[19,20]. Worldwide, there are an estimated 100 million clinical cases per year, of which around 5% progress to dengue shock syndrome or dengue haemorrhagic fever. We detected DENV-1 and 3 in this study, in keeping with previous reports of genotypes 1–3 on the African continent. The presence of both genotypes highlights the potential risk of dengue shock syndrome and dengue haemorrhagic fever in sequentially infected people, and in keeping with this, one case had evidence of haemorrhage at presentation. YFV is endemic in Africa and South America, and severe disease with haemorrhage occurs in up to 12% of those infected[21]. An effective live vaccine exists for YFV and has recently been used to control outbreaks in Brazil and DRC. National vaccination for YFV started in Uganda in 2023.

We detected two rhabdoviruses in this study; LDV, a potential pathogen and Adumi virus, a likely contaminant. Rhabdoviruses are highly divergent and ubiquitous in host range, infecting plants, invertebrates and vertebrates. Human disease is well-described; rabies lyssavirus causes fatal neurological infection, and Chandipura virus causes acute neurological disease. Other, less well-described members have also been proposed as human pathogens, including Bas Congo virus that was associated with a single outbreak of haemorrhagic disease in the DRC. Additionally, studies have also identified rhabdoviruses in blood from healthy asymptomatic individuals[22,23]. LDV is an emerging pathogen and has only two historical documentations; in a 10-year-old febrile patient with hepatosplenomegaly in Senegal in 1965 (sequenced in 2015[12]) and in a 47-year-old shipyard worker with fever and severe neurological symptoms in Wales in 1969[24], who had off-loaded a ship inbound from Nigeria (identified by serology). LDV belongs to the genus *Ledantevirus* and closely related species have been found in rodents and bats. Neurological pathology has been described in rodents infected experimentally with the virus[24]. In our study, we demonstrated an immune response and the development of neutralising antibodies in the infected individual and a household contact who had similar symptoms at the time.

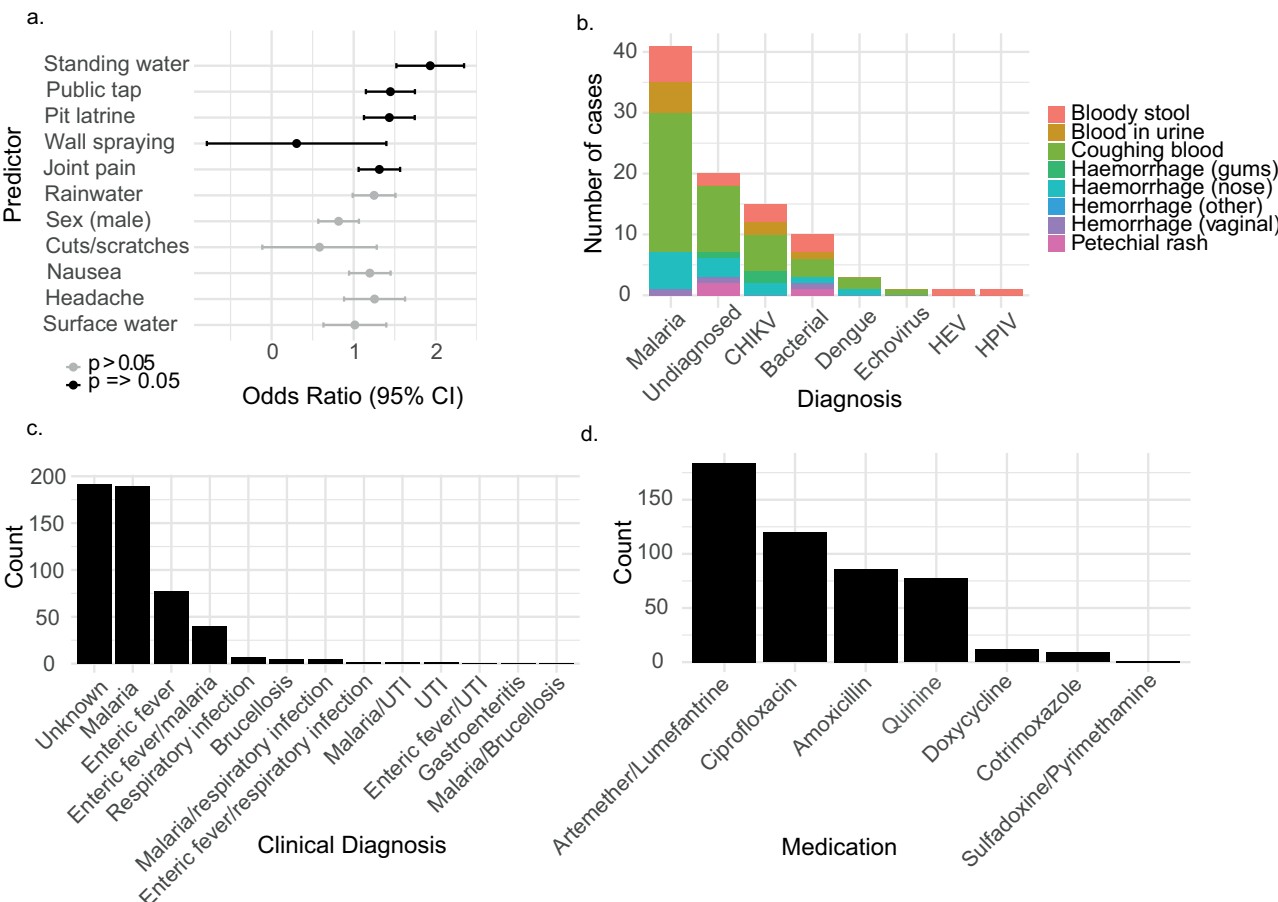

**Fig. 5 | Clinical and environmental variables associated with viral infection.**
**a** Multivariable logistic regression analysis (two-sided) of factors associated with viral infection for $n = 1281$ patients. Environmental and clinical history variables reaching $p = <0.05$ were included in a stepwise analysis. 95% confidence intervals are shown as error bars. Factors found to reach statistical significance are shown in black, while those not reaching significance are shown in grey. **b** Diagnoses associated with haemorrhage and fever. Cumulative totals of haemorrhagic symptoms in relation to the underlying diagnosis. Presentation with the viral hae-morrhagic fever viruses CCHFV, RVFV and YFV was not associated with haemor-rhage in the AFI cohort. **c** Initial clinical diagnosis in cases of confirmed viral infection. **d** Antimicrobial therapies administered in cases of confirmed viral infection.

Vaccine-preventable diseases remain common in Uganda, even for viruses covered under routine immunisation, indicating insuffi-cient coverage to generate herd immunity. Uganda has an active Expanded Programme of Immunisation (EPI) against measles, HBV and rotavirus. Vaccine coverage is lower in some neighbouring countries, and this may increase the risk of imported infection. Measles vaccine in Uganda has recently been increased from a one to a two-dose regimen and this is likely to improve the incidence of infection in future[25].

We aimed to identify viruses associated with AFI during suspected outbreaks of yellow fever and Rift Valley fever across Uganda. We detected several viral pathogens in such patients, highlighting that clinical presentation and serological assays are insufficient to identify numbers of cases for national reporting purposes. Flavivirus serology, in particular, may result in an over-estimate of reported ZIKV cases, due to cross-reactivity during DENV infection. The detection of CCHFV in this study outside the context of a known outbreak also confirms that this infection is under-estimated in Uganda, as has been found in other recent studies[26].

Using the extensive clinical and epidemiological data recorded for the whole AFI cohort, we performed a statistical risk analysis to ascertain clinical predictors and environmental drivers of viral infec-tions reported through the initial screening of the cohort assimilated with our own mNGS investigation. Our findings highlight key environmental and clinical factors that may influence the risk of viral infections, particularly arboviruses. Strong positive associations with exposure to standing water, and the use of public taps, and a negative association with spraying walls with insecticide underscores the importance of vector control in mitigating disease risk. The sig-nificance of joint pain as a clinical symptom of CHIKV infection pro-vides valuable insight into the likely burden of this virus (or related alphaviruses) in Uganda. The overuse of antimicrobial therapies for viral infection highlights the need for better diagnostics and anti-microbial resistance stewardship.

There were several limitations to our study. Firstly, the high detection of viruses associated with respiratory and gastrointestinal infection in plasma highlights the high burden that these diseases carry in Uganda. Enteroviruses, in particular, cause a range of diseases, which include respiratory, cardiovascular, gastrointestinal and neuro-logical sequelae, many of which are as yet not clearly linked to specific subtypes within the *Picornaviridae* family. We and others have pre-viously described the Saffold virus from Uganda, a *Cardiovirus*, first isolated from a stool sample in 2007. It has been linked to sympto-matic diseases ranging from measles-like illness to meningitis. The inclusion of respiratory and stool samples (not available in this study) in future studies will likely reveal a much higher prevalence of infec-tion. Additionally, the inclusion of serological assays would increase the detection of infections outside the viraemic phase. We have likely

under-estimated the burden of viral disease in this study for these reasons.

Viral metagenomics, although a powerful and sensitive method for unbiased genomic discovery, is prone to limitations. The limit of detection using mNGS is around $10^5$ copies/ml for >95% coverage of the genome for SARS-CoV-2, HCV, EBOV and AAV2 and $10^4$ copies/ml for detection[27–30]. Caution must be practiced when interpreting NGS data to exclude artefacts, which arise from cross-contamination from the environment, sequencing reagents and index hopping[31]. In this study, we found a novel rhabdovirus in several patient samples, most likely as a result of contaminated blood tubes with a virus of mosquito origin. This highlights the need for careful interpretation of mNGS data, confirmation by other methods and the development of standards and validation methods to interpret experimental information.

Febrile illness is widespread in Uganda and a significant proportion of undiagnosed disease can be attributed to viral infections, several of which have pandemic potential and can cause high mortality and morbidity. Metagenomic next-generation sequencing is a powerful tool to investigate the viral aetiology of AFI in high-risk populations. While unlikely to be cost-effective for individual diagnostic use, when applied to syndromic surveillance, it can be applied to enhance lower-cost clinical diagnostics in different populations. The finding of multiple highly pathogenic and poorly described viruses in a multi-site study in Uganda highlights the depth of the unknown with regard to circulating human viruses and the risk for onward transmission in a rapidly changing world.

## Methods

### Patients and sampling
About 1281 patients were recruited prospectively with informed consent into the AFI study from three study sites in Uganda; Ndejje HC IV (Wakiso), St. Paul's HC IV (Kasese), and Adumi HC IV (Arua) between April 2011 and January 2013 (Fig. 1a). Participants aged 2–17 years (and those unable to provide written consent) required written consent from parents/legal guardians. In addition, those aged 7–17 years assented to participation. Inclusion criteria were age ≥2 years with (a) a fever lasting 2–7 days or (b) symptoms consistent with brucellosis or typhoid fever, as previously described[10]. Cases with clinical evidence of alternative diagnoses, such as otitis media were excluded. Samples were obtained at presentation (acute sample) and 14–21 days later (convalescent sample). Diagnostic assays including serology, blood culture and blood films were carried out to identify infection with malaria, typhoid, leptospirosis, rickettsiae, CHIKV, dengue virus (DENV), WNV, YFV and ONNV. About 210 patients remained undiagnosed and plasma from these patients were retrospectively tested using mNGS. In addition, 20 samples obtained from suspected YFV, RVFV and VHF virus outbreaks referred to diagnostic services at UVRI between April 2013 and July 2016 were also included. These samples were tested for YFV, WNV, DENV and ONNV by PCR and for YFV, WNV, DENV, ZIKV and CHIKV by IgM as part of yellow fever virus outbreak surveillance (Supplementary Data 3). Additionally, three samples were tested for EBOV, MARV and CCHFV as part of VHF surveillance and one for RVFV.

### Ethics statement
Ethical approval for the AFI study was granted by the UVRI Research Ethics Committee (GC/127/10/02/19) and the Uganda National Council for Science and Technology (HS767).

### Statistical analysis
Clinical, environmental and socioeconomic demographic variables prospectively recorded for the 1281 AFI patients were subjected to a contingency analysis for the outcome of viral infection determined by Part I diagnostic screening and Part II mNGS. Univariable analysis was performed through Chi-squared or Fisher's exact tests for all categorical variables and t-test or Mann–Whitney for continuous variables. Variables reaching statistical significance ($p < 0.05$) were included in a stepwise multivariable logistic regression analysis.

### Sample processing and mNGS
RNA was extracted from 200 μl of plasma using paramagnetic bead-based technology and treated with DNase. RNA was reverse transcribed, followed by dsDNA synthesis and NGS libraries were prepared using an LTP low-input Library preparation kit (KAPA Biosystems) as previously described[32]. Resulting libraries were quantified, and up to 21 libraries were pooled in an equimolar ratio for each run and sequenced on Illumina MiSeq platforms at UVRI and the CVR, resulting in a median of 1.3 million read pairs per sample (Supplementary Fig. 1a). Repeat confirmatory sequencing was carried out on a subset of samples.

### Bioinformatic analysis
Raw fastq files were searched for viral sequences using DIAMOND BLASTx (Supplementary Fig. 1b)[33]. De novo assembly was carried out using dipSPAdes[34] and contigs were filtered using DIAMOND BLASTx against the nr database. Viral hits detected with BLASTx were confirmed with BLASTn and mapped to the closest reference genome using Tanoti (github.com/vbsreenu/Tanoti). If a virus was detected in multiple samples, minimum cross-contamination thresholds were used to confirm the presence of unique viruses in each sample, including (a) a read threshold of at least 10 reads when samples from the same run contained the same virus at ≥50 reads, (b) at least 50% of mapped reads were unique and (c) these sequences did not cluster identically by phylogeny. Taxonomic assignments were made on the basis of a BLASTn e-score of at least 0.01. In an additional independent analysis using human viral pathogen genome sequences, we estimated that 28 continuous nucleotides is a reliable minimum sequence length required for the taxonomic assignment of viruses, well under the minimum sequence length in our samples (Supplementary Fig. 1c).

Multiple sequence alignments were generated using MAFFT v7.487 with local alignment and 1000 iterations[35]. Maximum-likelihood phylogenetic analysis was carried out for viruses with at least 10 percent genome coverage using IQ-TREE multicore version 1.6.12 with 1000 ultrafast bootstrap replicates using relevant reference sequences[36]. Substitution models mentioned in figure legends were best-fit models as reported by IQ-TREE. Uncorrected pairwise-distances were estimated using MEGA 10.0[37].

### Capture ELISA
Rhabdovirus glycoprotein genes excluding the transmembrane regions were amplified and cloned into the secretory mammalian expression vector pHLSec containing a C-terminal 6xHistidine tag[38,39]. Human embryonic kidney cells (HEK-293T) were transfected, and cell supernatant containing secreted recombinant glycoproteins was harvested at 48 h. ELISA plates were coated with rabbit anti-His antibody overnight and blocked with 2.5% BSA/PBS at 37 °C for 1 h. Harvested cell supernatant was added to the wells, followed by patient serum, both for 1 h at 37 °C. This was followed by HRP-conjugated goat anti-human IgG at room temperature for 1 h. Reactions were developed using TMB substrate, stopped with 0.16 M sulphuric acid and read on a spectrophotometer.

### Pseudotype neutralisation assay
The use of vesicular stomatitis virus (VSV) in which the glycoprotein sequence has been replaced (VSVΔGluc) to generate pseudotyped viruses has been described previously[40]. Rhabdoviral glycoprotein genes were cloned into the eukaryotic expression vector and transfected into HEK-293T cells, followed by infection with VSVΔGluc-VSV-G. For neutralisation assays, $1 \times 10^4$ HEK-293T cells were plated in a 96-well plate and incubated at 37 °C for 1 h. Four-fold serum dilutions were mixed with either VSVΔGluc-VSV-G or VSVΔGluc-LDV-G. About 50 μl per well of the pseudotype/serum

mixture was added to the cells and incubated for 24 h. Luciferase substrate was added, and luminescence was measured on a plate scintillation counter.

Further details are presented in Supplementary Methods.

## Reporting summary

Further information on research design is available in the Nature Portfolio Reporting Summary linked to this article.

## Data availability

Sequence read data generated in this study has been deposited in the SRA database under Bioproject PRJNA1143542. Assembled virus genome sequences are deposited in Genbank with accession numbers OQ077983-OQ780008. Alignments and trees are available at https://github.com/ecthomson/AFI/. Anonymised clinical metadata is available at https://www.uvri.go.ug/sites/default/files/files/AFI%20data%20set%20%28web%29_shared.xls.

## Code availability

Code is available at https://github.com/ecthomson/Contamination-Phylogeny https://doi.org/10.5281/zenodo.14860977 and https://github.com/ecthomson/Contamination-Filterhttps://doi.org/10.5281/zenodo.14860987 uploaded on February 12, 2025.

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

## Acknowledgements

This study was conducted with grant funding from the UK Medical Research Council (E.C.T., S.A. and H.J., MC_UU_00034/6) and Wellcome Trust (ECT; 102789/Z/13/A). We would like to thank the Uganda Ministry of Health, health centre staff and all study participants for their contribution to the study.

## Author contributions

E.C.T. and J.B. conceptualised the study. S.A., H.J., T.B., G.S.W., J.F.S., M.G.S. and J.G.S. performed the experiments. E.C.T., H.J., S.A., C.W. and V.B.S. performed data analyses. E.C.T. and S.A. wrote the first version of the manuscript. J.T.K., M.G.S., D.L.B., D.S., C.D., N.L., A.S.F., A.S., W.W.M., B.J.W. and P.K. provided additional experimental data, advice and study facilitation. R.D., P.N., H.B., B.K.K., S.B. and J.L. assisted with collating data and samples used in the study. All authors contributed to the final manuscript review and revision.

## Competing interests

The authors declare no competing interests
