## [Transparent Peer Review file · Nature Communications]

Uncovering the viral aetiology of undiagnosed acute febrile illness in Uganda using metagenomic sequencing

Corresponding Author: Professor Emma Thomson

Version 0:

Reviewer comments:

Reviewer #1

(Remarks to the Author)

The study by Ashraf et al reports on the results of viral infections identified via metagenomic sequencing for 230 patient samples with no diagnosis following routine screening from Uganda. The majority of samples (210) were part of a previously published study on Acute Febrile Illness; the original paper reported on the results of screening for a panel of pathogens. An additional 20 samples were from 2 outbreaks. The study identifies a number of expected viral pathogens (e.g., HIV, hepatitis and yellow fever virus) as well as several rare or unexpected ones (e.g., Le Dantec Virus). The study is broadly descriptive, part of a growing body of publications reporting on detection of pathogens via unbiased sequencing from large cohorts. That said, the study is generally well conducted, the samples come from a very large cohort with extensive pre-screening (already published) and does go beyond the basic reporting of sequencing results in some areas (e.g., confirming exposure to Le Dantec virus via ELISA and neutralization).

The description of the bioinformatics analysis is too brief given its importance to the findings and warrants more detail. There are also several additional things I would like to know / that would be required to fully understand and repeat the analyses.

- The statement that "A diagnosis threshold of 28 nucleotides was estimated as a reliable cut-off" is ambiguous. Please provide additional details on how exactly this was implemented. In particular, how does this relate to the 10 reads cut-off used for minimum cross-contamination filtering?
- A 10% completeness cut-off for phylogenetic tree reconstruction is likely to lead to spurious or uninformative clustering. Are the authors confident in the multiple sequence alignments for these trees?
- What depth of sequencing was achieved / required per sample for analysis?

There are several areas where the authors could have further developed the analysis. In particular, a lot of information is presented in table 1 but it is exclusively descriptive with no statistical analyses presented and, as such, it adds little to the interpretation of the source of infections or the added value of incorporating such data. Given the challenges of collecting such complex data on this scale, this feels like a missed opportunity to ask questions such as "to what extent does a recent mosquito bite increase the likelihood of an arthropod-borne infection" and "could presumptive diagnoses have helped to narrow the search space for etiological agents or were they often wrong", or similar. This is commented on in the discussion (lines 291-292) but no statistics are there to support this. Indeed, it would be challenging to work out from the format in which the data are presented which cases of acute viral infection were initially misdiagnosed.

I found the presentation of the outbreak samples too brief and fragmented to be able to appreciate the importance of these results and was left a little unclear on the motive for including them. Given the difference in reason for collection, it feels important to consider them as distinct from the AFI samples but information to permit this is lacking. For example, where were these outbreaks (in the same sites as the AFI collections or elsewhere)? What was suspected as the cause(s) of the outbreak? Of the 7/20 cases where a pathogen was identified are these all from one outbreak or both? It would be helpful to add information on outbreak associated cases to table 2 to make this more clear. Regarding what was identified, based on the remarks in lines 176-179 and 206-209 these were possibly suspected to be YFV and the study identified 2 cases of DENV, 2 YFV, and 1 RVFV. Additionally it seems that HEV, HAV, HBV & rhinovirus were each detected in 1 individual associated with the outbreaks, HIV in 2 individuals and HHV5 in 4 individuals. Given this heterogeneity and lack of a single dominant pathogen across the samples, what can we conclude about the designation of these as just 2 outbreaks and/or the utility of mNGS for investigating them?

The Le Dantec virus finding is interesting, and it is strengthened by the ELISA and neutralization assays performed. Is it possible to screen the entire cohort (or some substantial proportion) to more broadly understand how widespread / under-reported this virus is in Uganda?

Regarding data accessibility, has the data been made publicly available? I could not see any reference to a BioProject or other public repository accession ID. This should be done prior to publication if not.

A few additional minor comments on formatting and presentation:

- Citation style fluctuates in some parts of the document (e.g., line 219 & 329-331) - please standardize.
- Author affiliations are not in numerical order, please revise
- Table 1: why does the health center diagnosis % field sum to > 100? Were some individuals given more than one presumptive diagnosis?
- Figures 2 and 4 include many coverage plots that take up a lot of space for relatively little information and reduce readability of the phylogenetic trees. Perhaps these could be shown with many samples together (as in Figure S2a) or summarized (e.g., a scatter plot of coverage depth vs breadth).

Reviewer #2

(Remarks to the Author)

The authors present a concise report on deploying metagenomic next-generation sequencing (mNGS) to identify causes of otherwise undifferentiated acute febrile illness in Uganda, examining a cohort of 230 microbiology-negative samples (210 from a previously tested cohort + 20 from two known outbreaks of febrile illness). They were able to detect potentially relevant pathogens in 44 samples, including 7 cases of arboviruses and an unusual rhabdovirus (LDV), which the authors have separately found to be widespread in the region based on seropositivity. The study is generally well designed and clearly written, and represents a good example of deploying mNGS in the context of population-based surveillance. A major strength is the incorporation of confirmatory lab work, including ELISA, and a thorough effort to rule out unexpected positive findings (insect-specific virus contamination of blood collection tubes).

Major comments:

Line 112 - It wasn't clear to me from the methods how positives were identified, and how potential artefacts were detected/removed. There is some confusion between the main text and the supplementary data; the "diagnosis threshold of 28 nucleotides" referred to on line 112 is the minimum kmer length for unique identification of known species, as per Suppl Fig 1b. This is not a "diagnostic threshold" in the laboratory sense, but a per-kmer threshold for taxonomic assignment, as used in many standard applications. Eg. Kraken (Wood et al 2014) similarly uses kmers of 31bp for the same reason of taxonomic identifiability based on uniqueness. Taxonomic identifiability is completely different from a diagnostic threshold. As the authors mention in their discussion, readily identifiable pathogen sequence fragments often turn up in negative samples due to minute amounts of laboratory contamination (aerosolisation, kit contamination, reagents, beads etc) or bioinformatic artefacts like index hopping. How many kmers (per read and per sample) were considered sufficient to call a sample positive - surely more than one kmer? Why is there a set of thresholds for confirming multiple positives (lines 121-123) but apparently not for singletons? I found this quite confusing. It also sounds like a kmer based approach was used to reduce database contamination by non-identifiable sequences such as repeats, but it's not clear what the actual process for calling positives was here. Please clarify in as much detail as possible.

Were there any pathogen-identifiable reads in the controls (were there negative controls)? This is not mentioned in the methods but is critically important for any clinical metagenomics study.

What was the expected limit of detection? Was this the same for all pathogens, or were some more readily identifiable?

Lines 166-167 state that "ten percent reported haemorrhagic symptoms but none of these had evidence of a viral haemorrhagic fever (VHF) virus by mNGS." This is a fascinating finding. Can you expand on whether symptom information was elicited in the same way from all patients? Is it surprising that there is a complete disconnect between hemorrhagic symptoms and presence of relevant pathogens?

Minor comments:

- * Author order in ms doesn't match submission details - please harmonise.
- * Line 38 - it's a bit of a stretch to say that Uganda itself is a "hotspot for zoonoses". At least some of the listed viruses emerged - and continue to emerge - outside Uganda, including in nearby geographic regions with poorer public health systems, but are detected and reported in Uganda. Can this be rephrased to make the language more sensitive?
- * Line 39 - please standardise capitalisation of virus names, and reword this sentence to include the word 'virus' after each, otherwise it reads as though there are outbreaks of placenames. SUDV and EBOV are both species in the genus ebolavirus; please keep the listing taxonomically consistent.
- * Line 89 - "acute plasma" sounds strange, is this meant to be plasma collected from symptomatic patients?
- * Line 104 - were samples sequenced individually or multiplexed on the same flowcell? This is a critical point when considering the possibility of read misassignment. Please state the number of reads obtained per sample
- * Line 108 - capitalise DIAMOND and dipSPAdes as per refs, and cite at the mention of the software, not at the end of the line (as this suggests that ref 14 refers to the nr database, for example).
- * YFV - was vaccination history available for the patients with YFV sequences detected?
- Figure 2 and lines 116-117 - are the substitution models listed in the legend those reported as best by IQTREE? Please

clarify in the methods, and state the version of IQTREE used here.

* I couldn't see any reference to multiple sequence alignment software to generate the alignments that feed into the phylogenetic reconstruction. Please expand methods as needed to include these steps.

* Data availability: I couldn't find a statement. Short read data should be deposited in the SRA/ENA; it would also be good to see the sequences with sufficient genomic coverage to build trees, as well as the trees themselves, made publicly available eg through github for reproducibility.

Reviewer #3

(Remarks to the Author)

This manuscript is an interesting application of pathogen-agnostic metagenomic analysis to clinical samples from febrile populations in Uganda whom received no diagnosis after conventional testing. I applaud the authors for their energy and enthusiasm in pursuing mNGS in a low-to-middle income setting where these sensitive analyses can be difficult to conduct, particularly the identification of Le Dantec virus. In terms of noteworthy results that would be applicable to a general audience reading Nature Communications, I am not convinced that this study (while interesting on a small scale) is best suited for a general journal given that there are: 1) larger mNGS cohorts of febrile illness previously published in Uganda, Cambodia and elsewhere which while it is important that this work by Ashraf et al be published, it is less novel given its scale and findings for a general journal; and 2) the follow-up confirmatory testing in addition to some of the viral coverage maps make it difficult to assess some of the viral calls made by mNGS.

MAJOR COMMENTS:

As mentioned in terms of noteworthy results and how this study contributes further to the field, the identification and characterization of Le Dantec virus via PCR and Sanger sequencing is certainly a gold standard approach. I would like to see a similar level of characterization for the other mNGS hits particularly CCHFV and even dengue, which had exceedingly low coverage and no other follow-up testing. I would have liked to have seen a Clinical Validation section in the Methods for each pathogen. I see the flow diagram in Supplement Figure 1A but it was not clear that this was indeed done for each patient represented in the study (e.g. CCHFV). I understand the difficulty in finding patients, but if sera was available for sequencing, why were additional PCR testing not done on original samples? For dengue, convalescent serology for flaviviruses lacks specificity and would not be the ideal way to confirm dengue diagnoses. That aside, this is a descriptive study for a discrete portion of Uganda, where there are certainly unidentified pathogens of concern, but without a significantly larger cohort, better characterized results, or further genomic analyses to relate to a larger phylogenetic picture, my belief is that this report may be better suited for a specialty journal.

In terms of reproducibility, I was confused if you were using NebNext DNA or RNA library prep kits because it is not clear in the Methods. For viral mNGS in clinical samples such as sera, RNA library prep kits would ideally provide more sensitivity and would be an option for you to enhance the coverage on some of your lower quality hits. However, the reference you use for your library preparation is a Personal View article in the Lancet Microbe (ref 10) that is not a scientific article with a Methods section or even reference to a specific study. This is a major error to be corrected.

Lastly, this is descriptive on a small scale. If this is part of a larger cohort, are there any further analyses to be done to add more information to this manuscript? For example, is extensive demographic/risk/climate data available to create a logistic regression model to understand various virus families from your larger cohort, to understand the likelihood of having an unidentified pathogen vs those identified via conventional means? Aside from LDV identification, I do not know what other applicable information a general reader can take away from the study.

Minor Comments:

Line 63-64 should be caveated in terms of costs and sensitivity of mNGS in certain clinical samples

Line 179 - can you say that they were 'diagnosed by mNGS' given it is not a validated tool? Would a more appropriate clinical approach to say that 'of the 44 patients with pathogens identified by mNGS' and then actual diagnosis is made by pathogen-specific validated diagnostics...

MINOR:

Grammar and typos: line 30 states 'virures' instead of viruses in Summary

Line 42 'were described here' makes it sound like described in paper, not Uganda. Please change.

Line 48 - it is misattributed 'to' not 'as'

Line 159 - remove dash

Reviewer #4

(Remarks to the Author)

Version 1:

Reviewer comments:

Reviewer #1

(Remarks to the Author)

The authors have addressed all my comments.

Reviewer #2

(Remarks to the Author)

My comments have been addressed in full in the revised ms. I have a couple of queries on the additional material and minor comments below.

* Lines 195 and 354: 16% refers to a denominator composed of both population-level screening and outbreak-targeted sequencing, which may be misleading. It would be better to split these out, similarly to the preceding paragraph.

* Figure 5a: were outbreak samples and population samples combined for the statistical analysis - if so, it would make sense to include a variable for whether the sample was part of a known outbreak.

* Line 313 lists virome components associated with these samples: have complete genomes been deposited in GenBank?

* Pegivirus was detected in 43 samples, which suggests high prevalence at least in this cohort. This is a virus commonly observed in high-burden settings. Although by no means essential to this paper, it would be interesting to know whether presence of pegivirus was associated with any clinical symptoms and/or co-infection with clinically relevant viruses?

Typos etc:

* Figure 5 all panels - model variables etc are barely legible in the PDF, please check that the SVG files provided render properly in the final version. I had to go to the SVG file to read the labels.

* Line 197 refers to paraflu 3 whereas table 1 lists it as reprovirus 3 - please harmonise.

* Supplementary table 2 - some sample IDs have been converted to date format by Excel

* Supplementary tables 3 and 4 - PDF formatting runs across pages, please adjust column width to fit to one page wide or save as landscape.

Reviewer #3

(Remarks to the Author)

Thank you for your replies. All my concerns are addressed.

REVIEWER COMMENTS

Reviewer #1 (Remarks to the Author):

“The study by Ashraf et al reports on the results of viral infections identified via metagenomic sequencing for 230 patient samples with no diagnosis following routine screening from Uganda. The majority of samples (210) were part of a previously published study on Acute Febrile Illness; the original paper reported on the results of screening for a panel of pathogens. An additional 20 samples were from 2 outbreaks. The study identifies a number of expected viral pathogens (e.g., HIV, hepatitis and yellow fever virus) as well as several rare or unexpected ones (e.g., Le Dantec Virus). The study is broadly descriptive, part of a growing body of publications reporting on detection of pathogens via unbiased sequencing from large cohorts. That said, the study is generally well conducted, the samples come from a very large cohort with extensive pre-screening (already published) and does go beyond the basic reporting of sequencing results in some areas (e.g., confirming exposure to Le Dantec virus via ELISA and neutralization).”

We thank the reviewer for the summary of results and comments regarding the size of the cohort and extensive pre-screening. Regarding the comment about the study being “broadly descriptive”, we have thought carefully about how we can comment more widely on the burden of viral disease in Uganda, using results merged from the original paper and this one, and making use of the very large prospective cohort study on which the work is based. We have incorporated this now into our analysis, using the prospectively collected clinical metadata available from the study. We think that this has enhanced the paper and thank the reviewer for the comment.

“The description of the bioinformatics analysis is too brief given its importance to the findings and warrants more detail. There are also several additional things I would like to know / that would be required to fully understand and repeat the analyses.”

We have added more detail to the bioinformatic analyses in the methods, as suggested by the reviewer.

“The statement that “A diagnosis threshold of 28 nucleotides was estimated as a reliable cut-off” is ambiguous. Please provide additional details on how exactly this was implemented. In particular, how does this relate to the 10 reads cut-off used for minimum cross-contamination filtering?”

Yes, we acknowledge that this was unclear and have rephrased this statement and clarified how we make calls regarding taxonomic assignments. We assign virus identification to each sample, based on (1) Taxonomic cut-off, and (2) Cross-contamination threshold.

1. **Taxonomic assignment** of genomes was carried out on the basis of BLASTn results (and e score) following a BLASTx filtering stage. Phylogenetic analysis was subsequently carried out using ICTV guidelines and the published literature, as described below. We have added e-scores to **Supplementary Table 1** which indicate the reliability of the taxonomic assignment. In an additional analysis, we established that a minimum continuous 28 nucleotide contig is sufficient for taxonomic assignment of viruses, as shown in Supplementary Figure S1c. However, the contig lengths in this study are substantially longer than this (due to the use of 150 base pair, paired end

sequencing) and it was not used as a practical cut-off. The BLASTn e-score was the primary measure of reliability of each assignment. This has been clarified in the text.

2. **Cross-contamination threshold** A 10 read cut-off was a conservative approach used to exclude samples with very low numbers of reads for any virus if another sample in the run was also positive for the same virus (more than 50 reads) and was phylogenetically indistinct. This step is not related to taxonomic assignment but is used to address the risk of low-level cross-contamination at any stage of the process (including during sample aliquoting and pipetting, as well as index hopping). We didn't detect cross-contamination in the control samples, but we selected this as a conservative read threshold.

“A 10% completeness cut-off for phylogenetic tree reconstruction is likely to lead to spurious or uninformative clustering. Are the authors confident in the multiple sequence alignments for these trees?”

We are confident that the phylogenetic reconstructions are accurate – to provide the reviewer with reassurance, we have added further detail on the alignment methods and also the number of nucleotides for each sequence used in **Supplementary Table 1**. The size of genomes used is comparable to other phylogenetic studies in the literature, and to the size of PCR products that are commonly used for phylogenetic analysis. The two shortest genome sizes used in the phylogenetic analyses were 380 nucleotides and 571 nucleotides for genome sizes of 1.6kb and 5.3kb respectively. We agree that with lower coverage, there is a risk of uninformative clustering. However, there was sufficient genetic diversity in these results to include them. All other phylogenetic analyses were carried out with more than 1000 nucleotide coverage of genomes ranging from 6kb – 16kb in size. We feel that the risk of spurious results is unlikely, given that we used bootstrapping with 1000 replicates. Alignments were produced using MAFFT using the maxiterate 1000 option and then manually checked. These additional details have been added to the text. The alignments and trees are also now available at <https://github.com/ecthomson/AFI/>.

“What depth of sequencing was achieved / required per sample for analysis?”

We aimed for around 1 million reads per sample and the median total sequencing reads obtained per sample was 1.3 million reads – this has been added to the methods section as well as details on multiplexing of libraries for sequencing. Raw numbers of mapped reads have been added to **Supplementary Table 1**. The efficiency of sequencing could be improved using a target enrichment approach in future. However, with this approach there may be a risk of lowered sensitivity for novel or unexpected viruses for which probes have not been included in the enrichment design. We could also have increased the number of reads per sample sequenced, but were limited by resource constraints.

“There are several areas where the authors could have further developed the analysis. In particular, a lot of information is presented in table 1 but it is exclusively descriptive with no statistical analyses presented and, as such, it adds little to the interpretation of the source of infections or the added value of incorporating such data. Given the challenges of collecting such complex data on this scale, this feels like a missed opportunity to ask questions such as “to what extent does a recent mosquito bite increase the likelihood of an arthropod-borne infection” and “could presumptive diagnoses have helped to narrow the search space for etiological agents or were they often wrong”, or similar. This is commented on in the discussion (lines 291-

292) but no statistics are there to support this. Indeed, it would be challenging to work out from the format in which the data are presented which cases of acute viral infection were initially misdiagnosed.

We thank the reviewer for this comment. In order to address it, we obtained permission to analyse the prospectively collected clinical metadata associated with the undiagnosed samples and with the samples that were previously diagnosed using conventional diagnostic assays.

Demographics and viral outcome for the full cohort has previously been published. However, demographic and environmental factors associated with viral infection were not previously analysed statistically and so we have carried out an additional analysis to investigate these in more detail. To do this, we carried out a stepwise multivariate logistic regression analysis of factors associated with the clinical history, demographics and environment and the odds of acquiring a viral infection. These are now described in the revised paper. Importantly, several of these factors suggest that the burden of arboviruses is high (with exposure to standing water associated with a higher odds of infection and wall spraying with a lower odds). Also, fever with joint pain was associated with viral infection, largely in association with positive Chikungunya serology. This was not picked up by the clinical team as a discriminating clinical symptom. Finally, we found that in an unselected acute febrile illness cohort, viruses that are known to be associated with viral haemorrhagic fever were not usually associated with haemorrhage at presentation, except in one case of dengue (nose and gum bleeding). This means that it is highly likely that sporadic cases of VHF e.g. with CCHFV are commonly occurring undetected in the Ugandan population. This is a major concern, as such viruses may spread to other members of the community and to healthcare workers. It was a feature of the 2013-2016 Ebola outbreak in West Africa that the majority of patients presented with fever and symptoms such as diarrhoea that are far less specific for VHF-type infections. We are detecting a similar phenomenon in this prospective cohort. The absence of “classic symptoms” also presents a high risk of generating outbreaks locally, nationally and internationally.

Also, in order to address the comment about misdiagnosis, we have added detail to the paper in order to highlight for each viral diagnosis the clinical diagnosis and the use of antibiotic or antimalarial therapy. Viral infections were invariably misdiagnosed (we have added Figure 5 to illustrate this).

“I found the presentation of the outbreak samples too brief and fragmented to be able to appreciate the importance of these results and was left a little unclear on the motive for including them. Given the difference in reason for collection, it feels important to consider them as distinct from the AFI samples but information to permit this is lacking. For example, where were these outbreaks (in the same sites as the AFI collections or elsewhere)? What was suspected as the cause(s) of the outbreak? Of the 7/20 cases where a pathogen was identified are these all from one outbreak or both? It would be helpful to add information on outbreak associated cases to table 2 to make this more clear.”

The inclusion of outbreak samples was intended as part of the AFI design to evaluate the use of NGS to identify the aetiology of viral outbreaks. We have added more detail to the outbreak section as suggested. We felt that the inclusion of these samples in the study was justified, given the similarity of presentation with undiagnosed acute febrile illness (the samples were obtained from various sites across Uganda). The samples were obtained during known outbreaks of yellow fever and of Rift valley fever which had been confirmed by UVRI using

conventional PCR and serological assays using residual samples. The samples were additionally tested for flaviviruses, RVFV and VHFV using serology and PCR and results are summarized in **Supplementary Table 3**. Unfortunately, due to ethical constraints, we had less clinical information available on these samples.

“Regarding what was identified, based on the remarks in lines 176-179 and 206-209 these were possibly suspected to be YFV and the study identified 2 cases of DENV, 2 YFV, and 1 RVFV. Additionally, it seems that HEV, HAV, HBV & rhinovirus were each detected in 1 individual associated with the outbreaks, HIV in 2 individuals and HHV5 in 4 individuals. Given this heterogeneity and lack of a single dominant pathogen across the samples, what can we conclude about the designation of these as just 2 outbreaks and/or the utility of mNGS for investigating them?”

This is a helpful point, and we perhaps hadn't emphasized this aspect of our results sufficiently. The main interpretation of the heterogeneity of results in the outbreak samples is that while the pathogens known to be in the area were detected, the majority of suspected cases had an alternative diagnosis. This is important for several reasons. Firstly, when quantifying the extent of an outbreak, it is insufficient to rely on clinical presentation alone to generate the numbers of affected patients for national reporting purposes. Secondly, it highlights that several pathogens are circulating under the radar, some of which are high consequence. Thirdly, assuming a diagnosis during an outbreak may be unhelpful for patient care and could even put the patient at risk. This was recently an issue in Uganda during the Sudan virus outbreak in 2022, when cases of malaria and CCHFV were detected when testing contacts of patients with Sudan virus disease. This could also compromise patient safety – for example patients may be isolated in Ebola Treatment Centres (ETCs) unnecessarily, putting them at risk of a second infection. It highlights a gap in diagnostic provision to include point of care assays for VHFV in endemic settings. Also, we note that the serological assays were unhelpful in this context in distinguishing YFV from DENV. We have brought more of these observations into the revised paper.

“The Le Dantec virus finding is interesting, and it is strengthened by the ELISA and neutralization assays performed. Is it possible to screen the entire cohort (or some substantial proportion) to more broadly understand how widespread / under-reported this virus is in Uganda?”

Yes, we have done this - we assessed the seroprevalence of LDV on a cohort of samples collected from across the country. We found that it is highly prevalent in some parts of Uganda and almost absent in others. This is now published in <https://www.ncbi.nlm.nih.gov/pmc/articles/PMC11257405/> and is cited in the paper. However, this is the first report of Le Dantec in AFI patients in Uganda (the above paper cites the preprint of this one).

“Regarding data accessibility, has the data been made publicly available? I could not see any reference to a BioProject or other public repository accession ID. This should be done prior to publication if not.”

Yes – both the SRA and metadata have been submitted as Bioproject PRJNA1143542. This detail has been added to the methods section.

“A few additional minor comments on formatting and presentation:

Citation style fluctuates in some parts of the document (e.g., line 219 & 329-331) - please standardize.”

Amended

“Author affiliations are not in numerical order, please revise”

Amended

“Table 1: why does the health center diagnosis % field sum to > 100? Were some individuals given more than one presumptive diagnosis?”

Yes - sometimes there was more than one presumptive diagnosis for each patient.

“Figures 2 and 4 include many coverage plots that take up a lot of space for relatively little information and reduce readability of the phylogenetic trees. Perhaps these could be shown with many samples together (as in Figure S2a) or summarized (e.g., a scatter plot of coverage depth vs breadth).”

While the reviewer has given us many very helpful suggestions, on this point, we have a preference to retain these figures in the current format, as they show variation in distribution across the genome in different samples, and are hence more descriptive of our data than summary plots would be.

We thank the reviewer for the many helpful suggestions and hope that we have addressed these points carefully.

Reviewer #2 (Remarks to the Author):

“The authors present a concise report on deploying metagenomic next-generation sequencing (mNGS) to identify causes of otherwise undifferentiated acute febrile illness in Uganda, examining a cohort of 230 microbiology-negative samples (210 from a previously tested cohort + 20 from two known outbreaks of febrile illness). They were able to detect potentially relevant pathogens in 44 samples, including 7 cases of arboviruses and an unusual rhabdovirus (LDV), which the authors have separately found to be widespread in the region based on seropositivity. The study is generally well designed and clearly written, and represents a good example of deploying mNGS in the context of population-based surveillance. A major strength is the incorporation of confirmatory lab work, including ELISA, and a thorough effort to rule out unexpected positive findings (insect-specific virus contamination of blood collection tubes).”

We appreciate the above comment. We feel that one of the major strengths of the study is the population-based systematic approach and we want to showcase this approach so that researchers across Africa and health ministries consider this approach to detect circulating infections. While individual diagnosis of viruses causing acute febrile illness using NGS is unlikely to be economically viable, population-based surveillance can help physicians and policy makers to understand “what we are missing”. Also, we thank the reviewer for pointing out the unexpected positive finding issue (Adumi virus) - again, we wish to highlight this finding – although it is not particularly exciting in terms of the result, it is critical that as people start to use these NGS-based methods, that they carry out confirmatory testing and don’t assume that all viruses detected in samples are human pathogens.

Major comments:

“Line 112 - It wasn’t clear to me from the methods how positives were identified, and how potential artefacts were detected/removed. There is some confusion between the main text and the supplementary data; the “diagnosis threshold of 28 nucleotides” referred to on line 112 is the minimum kmer length for unique identification of known species, as per Suppl Fig 1b. This is not a “diagnostic threshold” in the laboratory sense, but a per-kmer threshold for taxonomic assignment, as used in many standard applications. Eg. Kraken (Wood et al 2014) similarly uses kmers of 31bp for the same reason of taxonomic identifiability based on uniqueness. Taxonomic identifiability is completely different from a diagnostic threshold. As the authors mention in their discussion, readily identifiable pathogen sequence fragments often turn up in negative samples due to minute amounts of laboratory contamination (aerosolisation, kit contamination, reagents, beads etc) or bioinformatic artefacts like index hopping. How many kmers (per read and per sample) were considered sufficient to call a sample positive - surely more than one kmer? Why is there a set of thresholds for confirming multiple positives (lines 121-123) but apparently not for singletons? I found this quite confusing. It also sounds like a kmer based approach was used to reduce database contamination by non-identifiable sequences such as repeats, but it’s not clear what the actual process for calling positives was here. Please clarify in as much detail as possible.”

Yes, we acknowledge this was not as clearly written as it should have been – and these points are also mentioned by reviewer 1. We have now clarified these issues in the manuscript. NB – we didn’t use a kmer based screening approach at all for this cohort, because we find that while this is a sensitive method, it lacks specificity – the calculation of the 28 base pair “threshold” was therefore not one that we used – contig sizes were always far longer and in general we confirmed most of our findings (where samples allowed) with confirmatory tests, including for lesser known viruses such as Le Dantec where we designed ELISA and neutralization assays.

As mentioned above, we used both taxonomic and cross-contamination thresholds and have now clarified this in the text.

1. **Taxonomic assignment** of genomes was carried out on the basis of BLASTn results (and e score) following a BLASTx filtering stage. Phylogenetic analysis was subsequently carried out using ICTV guidelines and the published literature. We have added e-scores to **Supplementary Table 1** which indicate the reliability of the taxonomic assignment. In an additional analysis, we established that a minimum continuous 28 nucleotide contig is sufficient for taxonomic assignment of viruses, as shown in Supplementary Figure S1c. However, the contig lengths in this study are substantially longer than this (due to the use of 150 base pair, paired end sequencing) and it was not used as a practical cut-off. The BLASTn e-score was the primary measure of reliability of each assignment. This has been clarified in the text.
2. **Cross-contamination threshold** A 10 read cut-off was a conservative approach used to exclude samples with very low numbers of reads for any virus if another sample in the run was also positive for the same virus and was phylogenetically indistinct. This step is not related to taxonomic assignment but is used to address the risk of low-level cross-contamination at any stage of the process (including during sample aliquoting and pipetting, as well as index hopping). We didn’t detect cross-contamination in the control samples, but we selected this a conservative read threshold.

“Were there any pathogen-identifiable reads in the controls (were there negative controls)? This is not mentioned in the methods but is critically important for any clinical metagenomics study.”

We ran a negative extraction control during sequencing run which did not contain pathogen reads. We agree that this is an important issue (and now would standardly use a larger number of routine negative control samples in sequencing runs). To add a layer of reassurance, we used all sequenced samples for cross-contamination detection purposes, as described in further detail in the methods section, with a conservative cross-contamination threshold. We also confirmed pathogen presence by alternative methods where possible (by PCR and serology), as described in further detail in the revised manuscript.

What was the expected limit of detection? Was this the same for all pathogens, or were some more readily identifiable?

The limit of detection in our hands with metagenomic sequencing is around 10^5 copies per ml for >95% coverage of the genome (for SARS-CoV-2, HCV, EBOV, and AAV2), and a log lower for detection rather than full coverage. We have added this information with relevant citations to the text. While the technical limit of detection is similar for all viruses, they reach different viral loads in serum and may also have varying viral load dynamics. The fact that we detected respiratory and gastrointestinal pathogens in serum samples is likely the tip of the iceberg – if we were to have had access to respiratory and GI samples, we would very likely have detected more of these pathogens. This is discussed in the Discussion section.

Lines 166-167 state that “ten percent reported haemorrhagic symptoms but none of these had evidence of a viral haemorrhagic fever (VHF) virus by mNGS.” This is a fascinating finding. Can you expand on whether symptom information was elicited in the same way from all patients? Is it surprising that there is a complete disconnect between hemorrhagic symptoms and presence of relevant pathogens?

Yes, we agree that this is a major finding. Symptoms were collected systematically and prospectively and therefore we are confident that the information is reliable. We have expanded the discussion on this point a bit more – in fact many cases of Ebola virus disease in West Africa presented with fever and diarrhoea rather than haemorrhage and we suspect that this type of presentation is the tip of a far larger iceberg. Our CCHFV-infected case also did not present with haemorrhage. This means that we need to change the way we think about how VHFs present clinically – haemorrhage is not always present and is also often a late feature of severe disease. Agnostic sequencing at a population level allows us to point out these issues.

We have now also looked at the wider cohort for evidence of haemorrhage in non-outbreak cases of viruses that may classically be associated with VHF illness (including yellow fever, RVFV, CCHFV and dengue) and presented these results in the text. There was one case of bleeding (nosebleed and gum bleeding in one of the dengue-positive cases from part 1 of the study) but all others were negative for these symptoms.

Minor comments:

“* Author order in ms doesn't match submission details - please harmonise.”

This has been corrected.

“Line 38 - it’s a bit of a stretch to say that Uganda itself is a “hotspot for zoonoses”. At least some of the listed viruses emerged - and continue to emerge - outside Uganda, including in nearby geographic regions with poorer public health systems, but are detected and reported in Uganda. Can this be rephrased to make the language more sensitive?”

Yes, we thank the reviewer and agree that this language could perhaps be misinterpreted. We have rephrased. We also agree that viruses are more likely to be picked up in Uganda than in some of the neighbouring countries and have added this point to the introduction.

“Line 39 - please standardise capitalisation of virus names, and reword this sentence to include the word ‘virus’ after each, otherwise it reads as though there are outbreaks of placenames. SUDV and EBOV are both species in the genus ebolavirus; please keep the listing taxonomically consistent.”

This has been amended to list virus species consistently according to ICTV guidance.

“Line 89 - “acute plasma” sounds strange, is this meant to be plasma collected from symptomatic patients?”

This has been clarified/amended.

“Line 104 - were samples sequenced individually or multiplexed on the same flowcell? This is a critical point when considering the possibility of read misassignment.

Please state the number of reads obtained per sample”

Thank you for pointing this out. We have now included extensive additional supplementary information on sequencing and multiplexing details in the methods section.

“Line 108 - capitalise DIAMOND and dipSPAdes as per refs, and cite at the mention of the software, not at the end of the line (as this suggests that ref 14 refers to the nr database, for example).”

This has been amended.

“YFV - was vaccination history available for the patients with YFV sequences detected?”

No vaccination history was recorded for the participants. However, none of the participants are likely to have received vaccine as YFV vaccination was included in the national immunization program only in June 2023 (although mandatory in visitors to the country for longer).

“Figure 2 and lines 116-117 - are the substitution models listed in the legend those reported as best by IQTREE? Please clarify in the methods, and state the version of IQTREE used here.”

This has been added to the methods section.

“I couldn’t see any reference to multiple sequence alignment software to generate the alignments that feed into the phylogenetic reconstruction. Please expand methods as needed to include these steps.”

This has been added to the methods section.

“Data availability: I couldn’t find a statement. Short read data should be deposited in the SRA/ENA; it would also be good to see the sequences with sufficient genomic coverage to build

trees, as well as the trees themselves, made publicly available eg through github for reproducibility.”

This has been added.

Reviewer #3 (Remarks to the Author):

“This manuscript is an interesting application of pathogen-agnostic metagenomic analysis to clinical samples from febrile populations in Uganda whom received no diagnosis after conventional testing. I applaud the authors for their energy and enthusiasm in pursuing mNGS in a low-to-middle income setting where these sensitive analyses can be difficult to conduct, particularly the identification of Le Dantec virus. In terms of noteworthy results that would be applicable to a general audience reading Nature Communications, I am not convinced that this study (while interesting on a small scale) is best suited for a general journal given that there are: 1) larger mNGS cohorts of febrile illness previously published in Uganda, Cambodia and elsewhere which while it is important that this work by Ashraf et al be published, it is less novel given its scale and findings for a general journal; and 2) the follow-up confirmatory testing in addition to some of the viral coverage maps make it difficult to assess some of the viral calls made by mNGS.”

When selecting Nature Communications for this submission, we considered that this paper would be of particular interest to readers in the call for “Health in Africa” call. This is because strategically, we feel that our paper highlights the need to enhance surveillance methods to detect the presence of pathogens of high consequence in African countries. This is critical, given the ever-present danger of outbreaks of high consequence infectious diseases, both locally in country, within the African continent and globally. We would like to use the opportunity to suggest that the infrastructure and resources for similar population based NGS studies could be repurposed in many countries following the COVID-19 pandemic, thereby providing critical resilience for pandemic preparedness. We are aware of some of the studies that the reviewer mentions (and feel that these have been very useful contributions) but we feel that ours has the advantage of being (1) a systematic and prospective recruitment of a large number patients with acute febrile illness (rather than an opportunistic look at residual samples) for population surveillance in an LMIC country, and (2) involved extensive pre-screening for other pathogens, and (3) included extensive ecological and clinical data that we have now analysed in further detail. We perhaps undersold this information in our first draft, and therefore have pointed out these advantages more explicitly in the enclosed manuscript.

We agree that the Le Dantec observation is important – this detection (and subsequent finding that the infection is likely widespread in Uganda) highlights the fact that poorly defined viruses are circulating in Uganda. However, we would like to highlight other key findings in the study. The detection of high consequence infectious diseases including the viral haemorrhagic fever viruses CCHFV, RVFV, and YFV circulating under the radar is of serious concern and reflects a substantial risk for the emergence of high consequence infectious disease outbreaks in Uganda and beyond. By incorporating the wider prospectively collected clinical data, we point out that infected patients did not present with the “classic” presentation of fever and haemorrhage, indicating that we need to consider these infections in people who do not have VHF symptoms in endemic settings. Detection of these viruses strengthens the case also for the need to develop vaccines for CCHFV and RVFV and new treatments for all three viruses. On the basis

of the findings presented in this study, we have recently been awarded two major grants to carry out national agnostic enhanced surveillance using next generation sequencing (NIHR - £2M) and to formally assess the seroprevalence and incidence of CCHFV across Uganda (EEID - \$5.5M).

Given the detection both of circulating high consequence viruses in undiagnosed people and poorly described viruses, we feel that the findings are relevant for readers is deserving of a journal with wide outreach, not just to a specialty audience, but multidisciplinary scientists, policy makers as well as industry to enable better diagnostics and interventions, in particular in Africa.

MAJOR COMMENTS:

“As mentioned in terms of noteworthy results and how this study contributes further to the field, the identification and characterization of Le Dantec virus via PCR and Sanger sequencing is certainly a gold standard approach. I would like to see a similar level of characterization for the other mNGS hits particularly CCHFV and even dengue, which had exceedingly low coverage and no other follow-up testing. I would have liked to have seen a Clinical Validation section in the Methods for each pathogen. I see the flow diagram in Supplement Figure 1A but it was not clear that this was indeed done for each patient represented in the study (e.g. CCHFV). I understand the difficulty in finding patients, but if sera was available for sequencing, why were additional PCR testing not done on original samples? For dengue, convalescent serology for flaviviruses lacks specificity and would not be the ideal way to confirm dengue diagnoses. That aside, this is a descriptive study for a discrete portion of Uganda, where there are certainly unidentified pathogens of concern, but without a significantly larger cohort, better characterized results, or further genomic analyses to relate to a larger phylogenetic picture, my belief is that this report may be better suited for a specialty journal.”

For high consequence and putative emerging viruses, we attempted to confirm all diagnoses where possible by PCR, and/or serology and have added this detail to Supplementary Tables 1 and 3 and clarified this in the manuscript. We did not do this for more routinely expected viruses (e.g. measles, HAV, HAV, HIV).

For flaviviruses, we found that there was a degree of cross-reactivity and lack of sensitivity by PCR and serology, particularly for DENV detection. For the YFV cases, PCR was positive in both cases, IgM was positive in one and negative in the other. Dengue was detected in cases OB2 and OB3, but the DENV PCR was negative while the DENV IgM was negative or equivocal respectively. Interestingly, both samples were IgM positive for Zika. This highlights that flavivirus serology is highly cross-reactive and lacks sensitivity as has been described before. This highlights an important role for NGS in diagnosis of flaviviruses.

For the CCHFV sample (138-2), we performed qPCR in the VHF lab at UVRI and obtained a borderline Ct value of 41. We also contacted the patient to check on their condition and for convalescent serology, but they had moved to another country and was unavailable to attend for further sampling (NB – the convalescent sample was an integral part of the prospective study design, but follow-up was not always possible in every case). We are confident in the taxonomic identification of the virus, we obtained coverage of 380, 571, 1910 nucleotides from S, M and L segments respectively, of the CCHFV virus with an e-score of 1.00e-152, 3.00e-48, 0 – importantly these also clustered phylogenetically with other viruses from the same region. There was no other sample sequenced in the same laboratory with CCHFV detected, making cross-

contamination highly unlikely/impossible. Validated PCR-based diagnostics for CCHFV detect genomic regions much shorter than those that we obtained by NGS. We are therefore confident that this diagnosis was robust.

“In terms of reproducibility, I was confused if you were using NebNext DNA or RNA library prep kits because it is not clear in the Methods. For viral mNGS in clinical samples such as sera, RNA library prep kits would ideally provide more sensitivity and would be an option for you to enhance the coverage on some of your lower quality hits. However, the reference you use for your library preparation is a Personal View article in the Lancet Microbe (ref 10) that is not a scientific article with a Methods section or even reference to a specific study. This is a major error to be corrected.”

Yes, we used RNA library prep kits and the methods have been revised to make this clearer.

We apologise for the referencing error. The correct reference the methods should point to is reference 12 (Jerome *et.al.*) and this has been corrected.

“Lastly, this is descriptive on a small scale. If this is part of a larger cohort, are there any further analyses to be done to add more information to this manuscript? For example, is extensive demographic/risk/climate data available to create a logistic regression model to understand various virus families from your larger cohort, to understand the likelihood of having an unidentified pathogen vs those identified via conventional means? Aside from LDV identification, I do not know what other applicable information a general reader can take away from the study.”

We feel that the “descriptive” comment is perhaps a little unfair, in that this was a large and challenging prospective cohort study conducted in an LMIC and involved the recruitment of 1281 patients with the collection of detailed clinical metadata. We acknowledge that we may not have taken full advantage of the extensive metadata available, and we thank the reviewer for the suggestion to do further analysis which we now include.

We agree that the LDV finding is important. We are also of the opinion that the circulating high consequence viruses RVFV, CCHFV, YFV and DENV are of critical importance (we think this is in fact a more important observation) – not least because they mostly did not present “typically”. This means that we cannot rely on haemorrhagic fever to be present in such infections occurring sporadically in the community. They represent a substantial risk to act as the origin of larger outbreaks that can pose a significant risk locally and internationally.

This study demonstrates the values of population-based syndromic sequencing for enhanced surveillance in LMICs. As mentioned above, while individual sequencing is unlikely to be cost-effective, population sequencing can highlight public health risks, opportunities for public health intervention, diagnostic gaps which routine diagnostics can be implemented and research gaps – for example to develop vaccines and treatments for diseases with extremely high associated mortality (such as RVFV and CCHFV).

“Minor Comments:

Line 63-64 should be caveated in terms of costs and sensitivity of mNGS in certain clinical samples”

We are not suggesting that all clinical samples should be tested by NGS. However, testing in clinics is often limited to a very small number of pathogens (in some clinics, only for malaria),

and there are a high number of infections that remain undiagnosed in East Africa on a day-to-day basis. These pathogens may be of very high consequence – as highlighted by this paper – with substantial risks not just to affected patients but for onward transmission in affected communities and also healthcare workers of highly-transmissible pathogens with a mortality rate of more than 50% (as recently observed in Rwanda for example, where a large Marburg virus outbreak was only identified after 6 healthcare workers in an intensive care unit of two local hospitals died). In this scenario, it is of value to identify pathogens circulating under-the-radar and have reflected this in the text as ‘population-based syndromic surveillance by mNGS’. Recommendations by ministries of health can then base improvements to their diagnostic repertoire based on such prospective, population-based syndromic enhanced surveillance. It is highly likely that such surveillance would assist health authorities to identify high consequence circulating viruses in parts of the country where such infections might not be suspected.

“Line 179 - can you say that they were 'diagnosed by mNGS' given it is not a validated tool? Would a more appropriate clinical approach to say that 'of the 44 patients with pathogens identified by mNGS' and then actual diagnosis is made by pathogen-specific validated diagnostics...”

Yes, we agree with this point. NGS is not yet a validated tool in most countries and contexts (there are some high-income countries that have done this e.g. in the UK (UKHSA and UCLH), the USA (UCSF) and Australia (Melbourne)) – but we have not done this in Uganda (yet). To address this point, we have replaced ‘diagnosed’ with ‘with a viral genome identified’ in the text.

MINOR:

Grammar and typos: line 30 states 'virures' instead of viruses in Summary

Thank you - this has been corrected

Line 42 'were described here' makes it sound like described in paper, not Uganda. Please change.

This has been corrected

Line 48 - it is misattributed 'to' not 'as'

This has been corrected

Line 159 - remove dash

This has been corrected

Reviewer #4 (Remarks to the Author):

“I co-reviewed this manuscript with one of the reviewers who provided the listed reports. This is part of the Nature Communications initiative to facilitate training in peer review and to provide appropriate recognition for Early Career Researchers who co-review manuscripts.”

We thank the reviewers for further comments which are addressed as below.

* Lines 195 and 354: 16% refers to a denominator composed of both population-level screening and outbreak-targeted sequencing, which may be misleading. It would be better to split these out, similarly to the preceding paragraph.

We agree with the point but have a preference to keep line 195 as it is, as it continues on to describe all categories of viruses, not just VHFVs. The distribution of samples between AFI and outbreak have however been clarified and specified in further sections of the results. We have also modified a line in the discussion to reiterate the distribution for VHFVs.

* Figure 5a: were outbreak samples and population samples combined for the statistical analysis - if so, it would make sense to include a variable for whether the sample was part of a known outbreak.

Risk analysis (statistical analysis) was performed for the full AFI cohort of 1281 patients. No outbreak samples were included in this analysis.

* Line 313 lists virome components associated with these samples: have complete genomes been deposited in GenBank?

Raw read data for human symbionts and Adumi virus have been deposited to Genbank as metagenome assemblies under Bioproject PRJNA1143542. Full genomes for human pathogens have been deposited via BankIt and accession numbers have been added to the manuscript.

* Pegivirus was detected in 43 samples, which suggests high prevalence at least in this cohort. This is a virus commonly observed in high-burden settings. Although by no means essential to this paper, it would be interesting to know whether presence of pegivirus was associated with any clinical symptoms and/or co-infection with clinically relevant viruses?

Thank you for the interesting suggestion. We carried out an analysis to see if there were any associations with pegivirus infection but did not find any significant positive associations with clinical symptoms or with co-infections.

Typos etc:

* Figure 5 all panels - model variables etc are barely legible in the PDF, please check that the SVG files provided render properly in the final version. I had to go to the SVG file to read the labels.

We have increased the font size in Figure 5 – if there are still difficulties, this can be supplied in alternative formats if required.

* Line 197 refers to paraflu 3 whereas table 1 lists it as reprovirus 3 - please harmonise.

This has been corrected in table 1

* Supplementary table 2 - some sample IDs have been converted to date format by Excel

Thank you for spotting this. We have reformatted supplementary data to reflect correct entries.

* Supplementary tables 3 and 4 - PDF formatting runs across pages, please adjust column width to fit to one page wide or save as landscape.

These data are now submitted as supplementary excel files with the manuscript.